# Insights into adenosine A$_{2A}$ receptor activation through cooperative modulation of agonist and allosteric lipid interactions

**Agustín Bruzzese**[1,2,3], **James A. R. Dalton**[1,2,3]*, **Jesús Giraldo**[1,2,3]*

**1** Laboratory of Molecular Neuropharmacology and Bioinformatics, Unitat de Bioestadística and Institut de Neurociències, Universitat Autònoma de Barcelona, Bellaterra, Spain, **2** Unitat de Neurociència Traslacional, Parc Taulí Hospital Universitari, Institut d'Investigació i Innovació Parc Taulí (I3PT), Institut de Neurociències, Universitat Autònoma de Barcelona, Bellaterra, Spain, **3** Instituto de Salud Carlos III, Centro de Investigación Biomédica en Red de Salud Mental, CIBERSAM, Spain

* James.Dalton@uab.es (JARD); Jesus.Giraldo@uab.es (JG)

**Data Availability Statement:** All relevant data are within the manuscript and its Supporting Information files. The manuscript contains Molecular Dynamics (MD) simulations. Details of

## Abstract

The activation process of G protein-coupled receptors (GPCRs) has been extensively studied, both experimentally and computationally. In particular, Molecular Dynamics (MD) simulations have proven useful in exploring GPCR conformational space. The typical behaviour of class A GPCRs, when subjected to unbiased MD simulations from their crystallized inactive state, is to fluctuate between inactive and intermediate(s) conformations, even with bound agonist. Fully active conformation(s) are rarely stabilized unless a G protein is also bound. Despite several crystal structures of the adenosine A2a receptor (A2aR) having been resolved in complex with co-crystallized agonists and G$_s$ protein, its agonist-mediated activation process is still not completely understood. In order to thoroughly examine the conformational landscape of A2aR activation, we performed unbiased microsecond-length MD simulations in quadruplicate, starting from the inactive conformation either in apo or with bound agonists: endogenous adenosine or synthetic NECA, embedded in two homogeneous phospholipid membranes: 1,2-dioleoyl-sn-glycerol-3-phosphoglycerol (DOPG) or 1,2-dioleoyl-sn-glycerol-3-phosphocholine (DOPC). In DOPC with bound adenosine or NECA, we observe transition to an intermediate receptor conformation consistent with the known adenosine-bound crystal state. In apo state in DOPG, two different intermediate conformations are obtained. One is similar to that observed with bound adenosine in DOPC, while the other is closer to the active state but not yet fully active. Exclusively, in DOPG with bound adenosine or NECA, we reproducibly identify receptor conformations with fully active features, which are able to dock G$_s$ protein. These different receptor conformations can be attributed to the action/absence of agonist and phospholipid-mediated allosteric effects on the intracellular side of the receptor.

the simulations (input, scripts and topology files) are provided in S1 File in the Supporting Information. The trajectories have been deposited in a public repository with URL address https://osf.io/mfk52/?view_only=04520633598249f8a4a6dab5dab53ab6.

**Funding:** This study was supported in part by Ministerio de Ciencia, Innovación y Universidades (SAF2017-87199-R). The funders had no role in study design, data collection and analysis, decision to publish, or preparation of the manuscript.

**Competing interests:** The authors have declared that no competing interests exist.

## Author summary

Unbiased microsecond-length MD simulations of the adenosine A2a receptor (A2aR) were performed in quadruplicate, starting from the inactive conformation either in apo or with bound agonists: adenosine or NECA, each of them embedded in two different homogeneous phospholipid membranes. Different intermediate or active receptor conformations were found depending on the presence/absence of bound agonist and type of lipid environment. Exclusively, in DOPG with bound agonist, we reproducibly identify receptor conformations with fully active features, which are able to dock G$_s$ protein. These different receptor conformations can be attributed to the action/absence of agonist and phospholipid-mediated allosteric effects on the intracellular side of the receptor. Dynamic structural data are key for the understanding of agonist-mediated GPCR activation simulated in realistic membrane environments.

## Introduction

G protein-coupled receptors (GPCRs) are heptahelical transmembrane proteins [1, 2]. In the simplest scheme, they mediate many physiological and pathological processes by transduction of signals across cellular membranes and exist in a conformational equilibrium between active and inactive forms [3, 4]. In the absence of ligands, some GPCRs exhibit basal activity, thought to be caused by their surrounding environment, which provides enough energy for the receptor to obtain an active state [5–8]. Only in the active state are GPCRs able to bind cytosolic proteins, such as G protein or β-arrestin, mediating different downstream signalling pathways from the same receptor [9, 10]. Moreover, GPCR signalling effects can be modified by the binding of endogenous or exogenous extracellular ligands [11, 12]. Ligand binding triggers conformational changes in the orthosteric site that are amplified into larger conformational rearrangements [6]. For these reasons, GPCRs are one of the major targets of current market drugs [13]. Thus, it is essential to deeply understand ligand-dependent (de)activation of GPCRs in order to suitably understand human physiology and expectedly perform more efficient drug discovery.

Advancements in the crystallization process of GPCRs (and membrane proteins in general) have enabled several studies to gain important insight into features of their activation process [14, 15]. Most GPCR crystal structures belong to the class A (or rhodopsin-like) family, which represents the majority (approximately 85%) of the total superfamily [16, 17], and are important pharmaceutical drug targets [17]. As a result, class A family members are typically the most studied and possess a number of highly conserved sequence motifs [18], which presumably play important functional roles in common signalling mechanisms shared by all family members [16, 19]. In particular, the crystallization of class A GPCR active structures has served as an important factor in their proposed activation mechanism, such as: rhodopsin [20], β2 adrenergic receptor (B2aR) [21, 22], M2 muscarinic receptor (M2R) [23], adenosine A2a receptor (A2aR) [24, 25], and μ-opioid receptor (μOR) [26]. From this structural data, the common and simplest activation mechanism for class A GPCRs involves the movement of transmembrane helix (TM) 3, TM5, TM6 and TM7 with respect to each other [5, 6, 16, 27, 28]. The importance of water for G protein-coupled receptors (GPCRs) has also been supported by recent crystallographic data [29, 30] and from different studies showing how ordered waters interact with residues that are important in disease states, binding of drugs, receptor activation, and signalling [31–34]. However, as it has been pointed out [5, 35], the communication between the orthosteric ligand binding-site and the cytoplasmic region of the receptor

responsible for transducer protein binding is loose because they are not rigidly coupled. This includes an additional difficulty to the molecular understanding of ligand-dependent GPCR (de)activation. Molecular dynamics (MD) is a suitable computational technique for studying GPCR flexibility in a membrane environment. MD simulations at atomic resolution can give information on specific molecular processes, including interactions of proteins with lipids, receptor-ligand binding, and receptor conformational change [36]. There have been numerous studies on how GPCRs activate and transmit their signals from the extracellular side through to G protein binding on the intracellular side [5, 6, 18, 37–39]. In particular, MD simulations have been successful in providing accurate molecular features of GPCR conformational space [36]. In general, the typical behaviour of class A GPCRs, when they are observed without bound G protein, is to fluctuate between inactive and intermediate(s) states without inducing the fully active conformation, even in the presence of an agonist [40–48]. Therefore, one of the current challenges with GPCRs is to understand the molecular basis for their agonist-mediated transition into the active state. In a recently published experimental study, two different phospholipids proved to act as negative and positive allosteric modulators of the β2-adrenergic receptor (β2AR): 1,2-dioleoyl-sn-glycerol-3-phosphoethyl (DOPE) and 1,2-dioleoyl-sn-glycerol-3-phosphoglycerol (DOPG), respectively, while 1,2-dioleoyl-sn-glycerol-3-phosphocholine (DOPC) acted as neutral [49]. Consequently, we previously performed a computational study that supported these results by stabilizing the active conformation of apo β2AR in MD simulations for several microseconds using a DOPG membrane. This supports the notion that phospholipids may be involved in the activation process of class A GPCRs [50]. In particular, anionic DOPG (Fig 1A) appears to play a critical role in the stabilization of the active state of β2AR by making several electrostatic protein-lipid interactions [49, 50]. Meanwhile, DOPC (Fig 1B) (a close relative of POPC and a net neutral lipid) allows slow destabilization of the β2AR active state [49, 50]. These effects have been shown to be mediated by H-bond formation (or lack thereof) between the protein and phospholipid headgroups [51]. Moreover, data gathered in a β2AR ligand binding study suggests that protein and membrane interplay improves interaction between protein and ligand and could be important for drug development [49, 52]. Likewise, our recent computational study on homologous cannabinoid receptor 1 (CB1) found compelling evidence for positive allosteric communication between bound CB1 agonists and DOPG membrane phospholipids, which appears to enhance the speed of receptor activation [53]. Still, an interesting question is how much different membranes might affect the activation process of other class A GPCRs and the functional effect of their agonists.

A2aR is a prototypical GPCR class A member, ubiquitously expressed in the body [54]. On the one hand, it binds endogenous adenosine, which regulates vasodilation, inflammation [55] and affects the central nervous system (CNS), inhibiting dopaminergic activity [56]. On the other hand, the inhibition of A2aR by molecules like caffeine leads to an increase in dopaminergic activity, which makes antagonists of A2aR an important target for treating Parkinson's [57] and Alzheimer's diseases [58, 59]. Despite current knowledge, to date, only one selective A2aR agonist (Regadenoson) has gained FDA approval, as well as an antagonist (Istradefylline), which in combination with levodopa is used for the treatment of Parkinson's disease in Japan [58, 59]. A2aR is an ideal system to study GPCR activation processes because it has a high propensity for binding lipid allosteric modulators [60], is a well-known drug target for several agonists and antagonists, and has been the focus of recent NMR studies [42, 61–64]. These studies show that, upon the addition of a full agonist, A2aR activation follows outward movements of TM5 and TM6 (including rotation in the latter), an inward shift of the intracellular part of TM7, and a vertical translation of TM3 [42, 61–64]. Also, A2aR has received a lot of attention regarding ligand binding, lipid allosteric modulation and its activation process in MD simulations [44, 65–78] partly because it is a receptor that has been crystallized in three

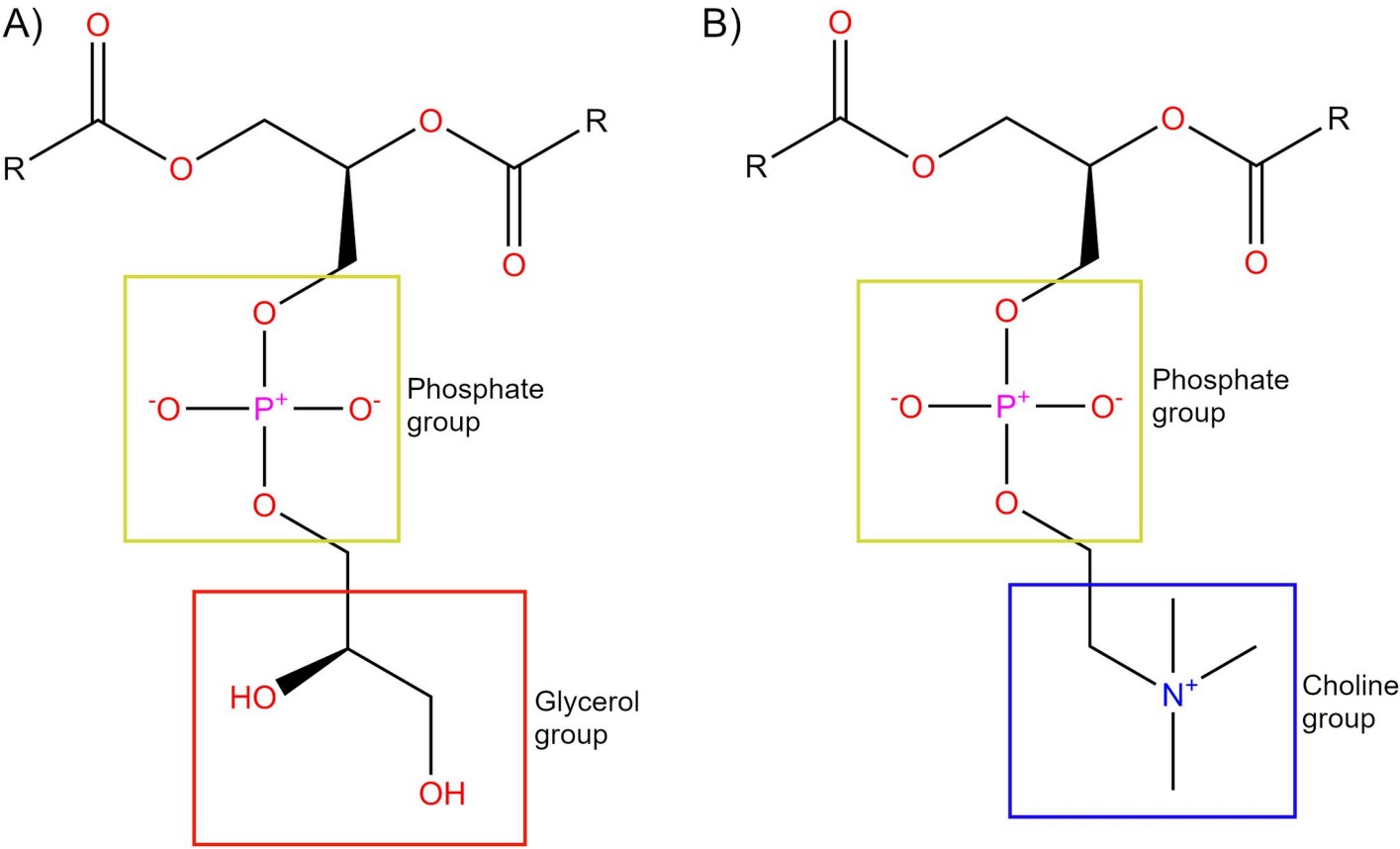

**Fig 1. Structural comparison of DOPG and DOPC lipid headgroups.** DOPG (A) headgroup is composed of a neutral glycerol group (red box) bound to the phosphate group (yellow box). DOPC (B) headgroup is composed of a positively-charged choline group bound (blue box) to the phosphate group (yellow box). DOPC and DOPG contain identical 18:1 fatty-acid chains (R).

distinct conformational states: inactive (in the presence of an antagonist or inverse agonist) [29, 63, 79–91], intermediate (in the presence of an agonist) [92–95], and active (in the presence of an agonist plus modified or native stimulatory G proteins) [24, 25] (S1 Table).

These crystal structures can constitute a reference point for comparison between active, intermediate, and inactive states of A2aR during MD simulations. Such published MD simulations usually share a common protocol. For example, intracellular co-crystallized molecules such as G$_s$ protein and activating nanobody are first removed (e.g. [44, 68, 71, 76]) and then performed within a homogeneous membrane consisting of 1-palmitoyl-2-oleoyl-sn-glycerol-3-phosphocholine (POPC) lipids (e.g. [65–76, 78]). This is due to the relative abundance of this particular phospholipid in healthy mammalian membranes [96, 97]. In this context, previous MD simulation data has shown that the structural motions of A2aR may depend on its phospholipid environment [65] as well as on cholesterol, which has been shown to enhance A2aR activation and signalling [77, 78]. On the other hand, recent MD studies have obtained results on the important role of ligand binding in A2aR activation [44, 67, 69, 70]. In addition, A2aR has been described as a weakly coupled system, due to the fact that binding of an agonist does not guarantee that the active state is reached under experimental conditions [41] and ligand binding may not be an exclusive driving force for achieving full receptor activation [44, 67, 69, 70, 78]. Thus, to our knowledge, the full agonist-mediated transition of A2aR from the

inactive form towards the active state has not yet been described with unbiased MD simulations and it remains unclear how agonists select or induce the active form of A2aR [41].

Thus, previously mentioned data indicates that ligand binding together with lipid membrane allosteric effects are important components of signal transduction that when systematically studied can lead to a deeper understanding of class A GPCR function, which could potentially lead to the development of more effective drugs [70]. To this aim, we have chosen to study the agonist-mediated activation process of A2aR by performing quadruplicated unbiased microsecond-length MD simulations with bound agonists: adenosine or 5'-N-ethylcarboxamidoadenosine (NECA) in two different homogeneous lipid environments: DOPC or DOPG (Fig 1). These two phospholipids contain the same 18:1 fatty-acid chains so are ideally suited for determining the allosteric effect of different lipid head groups on the receptor. Our results suggest that anionic phospholipids with a neutral head group such as DOPG play an important positive allosteric role in the activation of A2aR, similarly to β2AR and CB1, and work in a synergistic fashion with bound agonist. These observations might open new lines of ligand-lipid cooperativity in GPCR behavioural modulation and signalling at a molecular level.

## Results

NECA is a highly potent synthetic agonist of A2aR [98], while adenosine is a weaker endogenous agonist. Instead of the 5'-hydroxymethylene group positioned at C4' of the ribose moiety in adenosine (Fig 2A), NECA contains an N-ethylcarboxamido group (S1 Fig). Upon docking adenosine (Fig 2A) or NECA (S1 Fig) into the inactive human A2aR crystal structure [29], both ligands re-form their observed crystal interactions, which include an H-bond with E169$^{45.53}$ and aromatic π−π stacking with F168$^{45.52}$, both of which are located in the second extracellular loop (ECL2) (residues appended with Ballesteros X(Y).ZZ numbering [99] as superscript, which indicates relative position ZZ along TM helix X or loop between helices X and Y) (Fig 2B and 2C, S1 Fig). In addition, both adenosine and NECA make H-bonds with N253$^{6.55}$, S277$^{7.42}$ and H278$^{7.43}$ located on TM6 and TM7, respectively. This process yields an inactive receptor structure [29], crystallized at high resolution, with bound adenosine or NECA in a binding pose consistent with that previously crystallized in the thermostabilized A2aR intermediate state [93] (Fig 2B and 2C, S1 Fig).

In order to prepare for studying the dynamics of the A2aR activation mechanism at a molecular level, we initially compared the following crystal structures: agonist-bound, G$_s$ protein-bound active state [25], the inactive state bound to inverse agonist ZM241385 [29], and the intermediate state bound to adenosine or NECA [93]. In brief, the most remarkable conformational differences between them are located in: i) TM3, TM5, TM6, and TM7 helices, and ii) hydrophobic receptor core (Fig 2) (S1 Fig). Both the intermediate and active crystals share common structural differences with respect to the inactive crystal structure. Firstly, TM3 is rotated and moved upwards (from intracellular to extracellular) in the intermediate and active crystals. On this basis, a shift of residue L95$^{3.43}$ correlates with TM3 upward axial movement (S2 Fig, S3 Fig) [16]. In conjunction, W246$^{6.48}$ located on TM6 also changes orientation. This residue is known as the "toggle switch" in class A GPCRs because it can differentially rotate between alternative receptor conformational states [18, 100, 101], including A2aR [44], in particular. However, in the respective crystal structures of A2aR (S2 Fig, S3 Fig), dynamic rotation of W246$^{6.48}$ (sidechain dihedral angle χ1: 285°) is not observed but instead shows a vertical displacement of 1.7 Å. Moreover, the distance between TM3 and TM7 is decreased, most noticeably between intracellular residues R102$^{3.50}$ and Y288$^{7.53}$, which are 5.0 Å closer in the active state, while the distance between TM3 and TM5 is increased, exemplified by residues

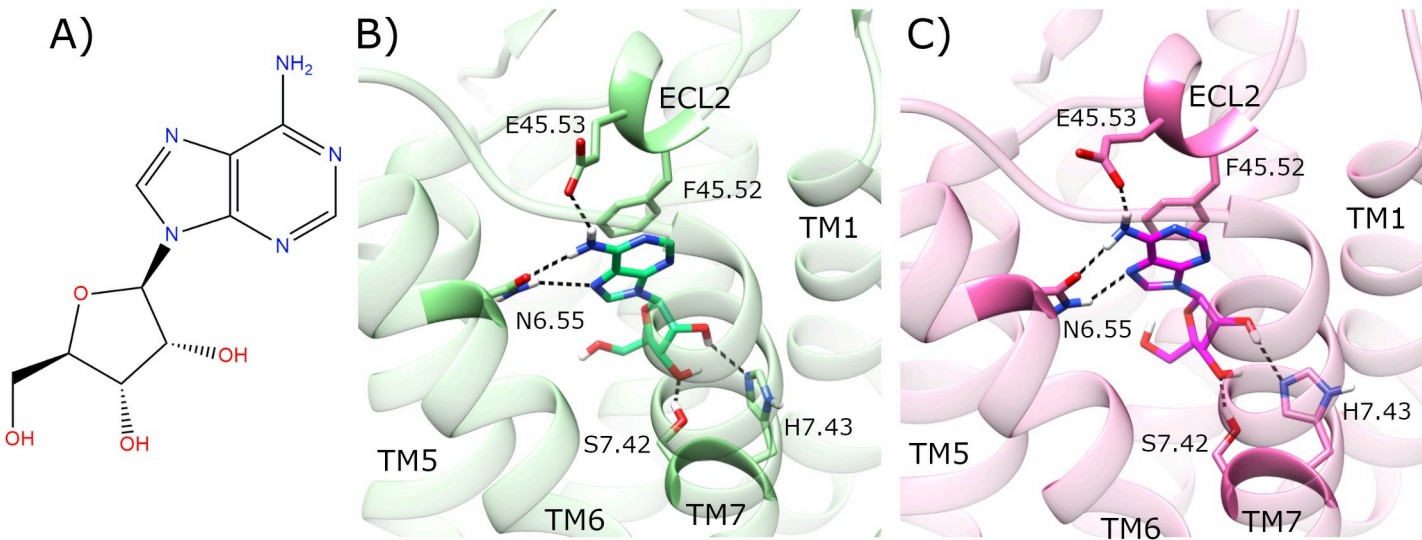

**Fig 2. Docking of adenosine in the inactive crystal structure of adenosine A2a receptor (A2aR).** A) Molecular structure of adenosine. Comparison of B) co-crystallized adenosine (lime) in agonist-bound A2aR crystal structure (PDB entry: 2YDO, light green), and C) docked adenosine (magenta) in the inactive crystal structure of A2aR (PDB entry: 4EIY, pink). Selected residues participating in ligand binding are displayed. ECL2 and TM helices 1, 5–7 are labelled.

R102$^{3.50}$ and Q207$^{5.68}$, which are 2.7 Å further away in the intermediate state (S2 Fig, S3 Fig) (S2 Table). The main difference between intermediate and active crystal structures can be observed in the intracellular distance between TM3 and TM6, and ionic-lock residues R102$^{3.50}$ and E228$^{6.30}$, in particular. Both crystals have TM6 oriented outwards compared with the inactive crystal structure. However, TM3-TM6 distance reaches 18.6 Å in the active, but only 9.9 Å in the intermediate crystal structure (S2 Fig, S3 Fig) (S2 Table).

In agreement with NMR studies [42, 61–64] and previous MD simulation data [44, 65–78], comparison of A2aR crystal structures points to TM3, TM5, TM6 and TM7 as key receptor-activating helices. Therefore, we base our quantitative MD analysis of A2aR activation/modulation on the following metrics: TM3-TM5, TM3-TM6, TM3-TM7 distances; W246$^{6.48}$ χ1 dihedral angle change; TM3 and ECL2 vertical (Z-axial) movement; root mean square deviation (RMSD) of transmembrane domain (TMD), ECL2, TM6 and L95$^{3.43}$. In addition, experimentally verified protein-agonist H-bonds involving residues N253$^{6.55}$, E169$^{45.53}$, H278$^{7.43}$, S277$^{7.42}$, as well as π-π stacking with F168$^{45.52}$, are monitored. In addition, in order to characterize the role of internal water networks in receptor activation, water-mediated interactions in the receptor core between residues pairs are also quantified.

## Apo A2aR embedded in DOPC membrane remains in inactive state

Over four independent MD simulations of 2.0 μs each of apo A2aR embedded in a DOPC membrane, ECL2 displays noticeable variable behaviour. In replicas #2 and #3, a conformational change occurs (at 400 ns and 1000 ns, respectively) where ECL2 moves downwards by 2–4 Å to form a lid over the orthosteric pocket (Fig 3A–3F). Over the course of this process, ECL2 reaches a maximum of 8.6 Å and 10.9 Å RMSD, respectively (S4 Fig). However, in replica #3, the loop moves back towards its original position, away from the orthosteric pocket. This shows alternative conformations of ECL2 are possible but not necessarily physically stable. Nevertheless, in replica #2, the loop remains in a downwards position partially covering the top of the orthosteric pocket (Fig 3A), albeit with conformational fluctuations (S4 Fig). This conformational change of ECL2 is possible in the apo state due to the lack of bound

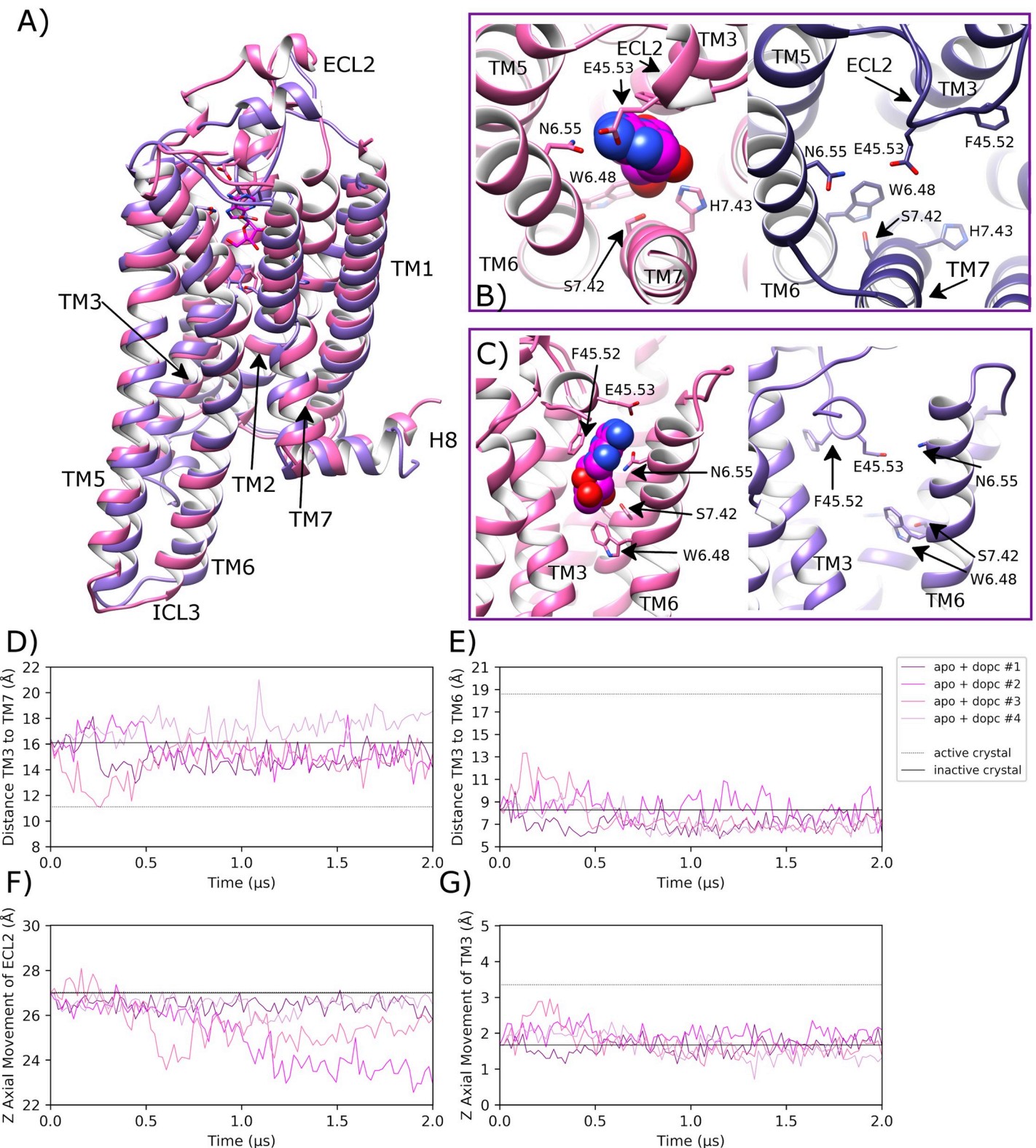

**Fig 3. Stabilization of the inactive conformation of apo A2aR in DOPC membrane.** A) Superposition of the inactive-state crystal structure of A2aR (PDB entry: 4EIY, pink) with docked adenosine and an MD-generated apo conformation achieved within a DOPC membrane (magenta, belonging to replica #2 from 1.7 μs) showing B) and C) selected residues delineating the orthosteric pocket. ECL2 and TM helices are labelled where applicable. D) Fluctuation of the distance between TM3-TM7 (from Cα atoms of R102$^{3.50}$ and Y288$^{7.53}$, respectively) during MD simulations, starting from the inactive crystal structure (PDB entry: 4EIY). E) Fluctuation of the distance

between TM3-TM6 (from Cα atoms of R102$^{3.50}$ and E228$^{6.30}$, respectively). F) and G) Vertical movement of ECL2 and TM3, respectively. MD simulations are performed in quadruplicate. Corresponding flat-lines are included to show the observed distance in the active (PDB entry: 6GDG) and inactive (PDB entry: 4EIY) A2aR crystal structures.

agonist and because the receptor remains inactive, thus generally maintaining the TM3-TM7 starting distance of 16.1 Å (Fig 3D). This allows the latter section of ECL2 to adopt a lower position between the extracellular ends of TM3 and TM7. However, in the other two replicas, ECL2 broadly maintains its original crystal structure position (average RMSD across replicas #1 and #4 of 2.4 Å (0.7 SD) (S4 Fig), which demonstrates ECL2 variability as previously discussed in class A GPCRs, especially in the apo state [102]. Despite the differences in observed ECL2 dynamics across replicas, little conformational change is observed in the receptor as a whole with an average RMSD of 2.2 Å (0.5 SD) (S5 Fig) or in key helices such as TM3 (average Z-axis displacement of 0.2 Å (0.6 SD), Fig 3G) and TM6 (average RMSD of 2.0 Å (0.7 SD), S6 Fig). Likewise, important residues such as L95$^{3.43}$ on TM3 and W246$^{6.48}$ on TM6 show no sustained conformational change across all four replicas (S7 Fig, S8 Fig). Likewise, the average distance between TM3 and TM6 (measured between ionic lock residues R102$^{3.50}$ and E228$^{6.30}$) is 7.8 Å (1.3 SD across all four replicas, which is similar to the inactive crystal structure distance of 8.2 Å (Fig 3E). In addition, the average distance between TM3 and TM5 is 14.9 Å (1.5 SD), which is indicative of a stabilized inactive receptor state across all four replicas (S9 Fig). Interestingly, TM7 appears to be one of the more flexible regions of the apo receptor as the distance between TM3 and TM7 (measured between R102$^{3.50}$ and Y288$^{7.53}$) is seen to vary more than other measured distances (Fig 3D, average distance of 15.8 Å (1.5 SD). For example, in replica #4, TM7 is seen to move outwards, increasing TM3-TM7 distance relative to the inactive crystal state, while in replica #3, TM7 temporarily moves inwards towards TM3 before reverting to its original state. Despite some of these observed differences, TM7 regularly revisits its inactive crystal conformation across all four replicas, suggesting this state is at least meta-stable and in line with the receptor remaining inactive (Fig 3D).

## Adenosine-bound A2aR in DOPC membrane reaches intermediate conformation

During quadruplicate MD simulations of inactive A2aR with bound adenosine embedded in a DOPC membrane, the ligand displays variable behaviour (Fig 4B). In replicas #3 and #4, adenosine achieves a stable binding mode in the orthosteric pocket between helices TM3, TM6 and TM7 (Fig 4), which is consistent with its crystallized pose in the A2aR intermediate crystal structure [93]. As a result, ligand RMSD does not exceed 4.5 Å over the course of either replica, finishing at 1.7 Å in replica #3 and 3.8 Å in replica #4 (Fig 4D). However, in replicas #1 and #2, adenosine changes its binding pose frequently without finding stability, in some cases moving up to a maximum of 8.9 Å. This is because adenosine is relatively small and is able to occupy an alternative space in the orthosteric pocket in between helices TM1, TM2 and TM7 (S10 Fig). However, this alternative binding pose is not stable as reflected by the high conformational fluctuation observed in these respective replicas (Fig 4E). As a result, in the replicas where adenosine achieves a stable binding pose (#3 and #4), residue E169$^{45.53}$ located on ECL2 makes a protein-ligand H-bond with an average occupancy of 5.4% (6.3 SD), while F168$^{45.52}$ makes a protein-ligand ring-stacking interaction with adenosine 12.3% (7.7 SD) of the time (S11 Fig). However, in replicas #1 and #2 where adenosine is less stable, the frequency of these interactions decreases to 2.9% (1.4 SD) and 8.4% (6.7 SD), respectively. Despite this difference across replicas, ECL2-ligand interactions assist in the maintenance of ECL2 near its original crystal conformation (average RMSD of 2.8 Å (0.8 SD), especially relative to apo A2aR in

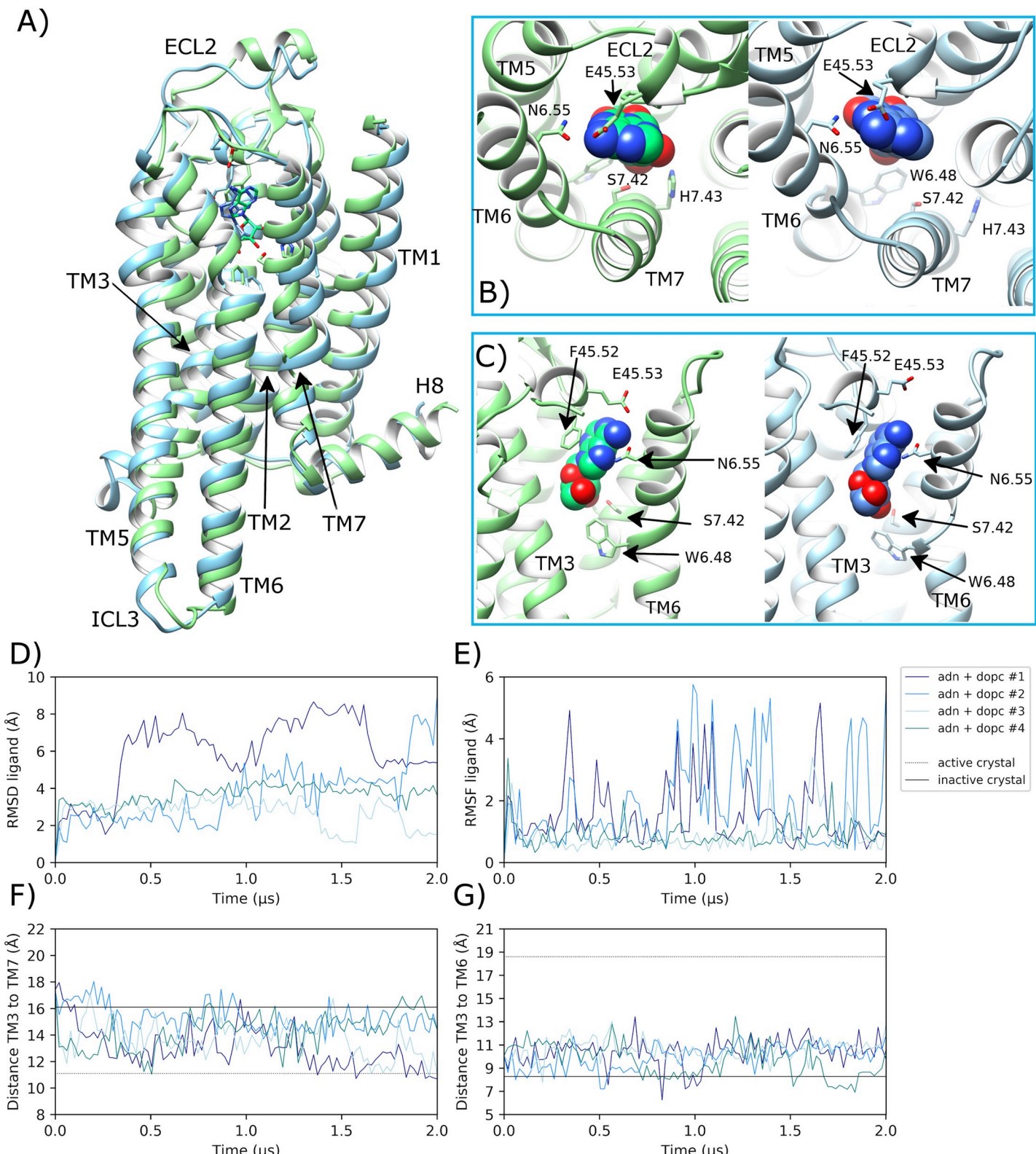

**Fig 4. Transition towards an intermediate conformation of adenosine-bound A2aR in DOPC membrane.** A) Superposition of the intermediate state crystal structure of A2aR (PDB entry: 2YDO, light green) and an MD-generated conformation achieved within a DOPC membrane (blue, belonging to replica #3 from 1.5 μs) bound to adenosine, showing B) and C) protein-agonist interactions in the orthosteric pocket with adenosine atoms displayed as spheres. ECL2 and TM helices are

labelled where applicable. D) RMSD of bound adenosine (ADN) (calculated with respect to initial docking pose). E) Conformational fluctuation (RMSF) of adenosine. F) Distance between TM3-TM7 (from Cα atoms of R102[3.50] and Y288[7.53], respectively) during MD simulations starting from the inactive crystal structure (PDB entry: 4EIY). G) Distance between TM3-TM6 (from Cα atoms of R102[3.50] and E228[6.30], respectively). MD simulations are performed in quadruplicate. Corresponding flat-lines show the observed distance in the active (PDB entry: 6GDG) and inactive (PDB entry: 4EIY) A2aR crystal structures.

DOPC (S4 Fig). This may also play a role in encouraging TM3 to move upwards into a more active conformation (ECL2 and TM3 are connected by a disulphide bridge) in replicas #3 and #4, where TM3 moves upwards (according to Z-axis) by an average of 0.8 Å (0.2 SD (S7 Fig)).

Regarding key protein-agonist interactions, N253[6.55] located on TM6 and S277[7.42] and H278[7.43] located on TM7 make H-bonds with adenosine with average occupancies of 92.1% (7.0 SD), 49.5% (25.2 SD) and 62.9% (16.1 SD) (S11 Fig), respectively, in replicas #3 and #4 where the ligand is more stable (conversely only 37.1%, 11.4% and 17.3%, respectively, for replicas #1 and #2). These interactions can be correlated with conformational changes in helices TM6 and TM7. For example, the distance between TM3 and TM7 decreases as TM7 moves inwards in all four simulations before 0.5 μs (Fig 4F). Although this conformational change is maintained until the end in only two replicas (#1 and #3, Fig 4F). On TM6, the rotameric state of W246[6.48] changes from g+ to trans in three out of four replicas after 0.25 μs (S8 Fig). This precedes the gradual separation of the intracellular end of TM6 away from TM3 by a maximum of 5.1 Å, which starts at a distance of 8.2 Å and increases up to 13.4 Å within 0.7 μs across these same three replicas (Fig 4G). However, as this conformational change of TM6 occurs mainly at its intracellular end, RMSD of TM6 shows only moderate change, reaching 4.0 Å in replica #4 (S6 Fig). Finally, conformational change is also observed in TM5 (3 out of 4 replicas) as it adopts a more outward orientation, up to 7.0 Å further away from TM3 (S9 Fig), mirroring changes in TM6. However, these changes in TM5 appear to be relatively unstable and fluctuate over time (S9 Fig) with an average distance from TM3 of 17.0 Å (1.9 SD) across all four replicas.

Taking these observed conformational changes together, the receptor achieves a full complement of intermediate-like crystal structure [93] features in three out of four replicas (#1, #3 and #4). This suggests partial receptor activation from the inactive state into a consistent intermediate conformation is readily achievable on microsecond timescales with bound agonist and membrane consisting of neutral phospholipids, although generation of a fully active receptor state is not. This is in general agreement with previous studies [44, 67, 69, 70]. Collectively, observed intermediate receptor features include a partial upwards translation of TM3 (i.e. not at the active crystal level), outward movement of TM5 and TM6 (only partial in case of TM6), and inward movement of TM7, resulting in closer interaction with TM3 (Fig 4). However, these conformational changes are only maintained consistently over 2.0 μs in replica #3, while in replicas #1 and #4 such intermediate-like features are more transient and coincide only briefly (specifically at 1.9 μs and 1.2 μs, respectively). This is likely due to the stable ligand binding-mode achieved in replica #3, including high H-bond occupancy with S277[7.42] compared to the other replicas (S11 Fig). The importance of this H-bond has been highlighted previously [103]. On the other hand, replica #2 fails to obtain an intermediate-like conformation of TM7, although some minor conformational changes are noted in TM3, TM5 and TM6, similar to other replicas. This lack of effect is likely due to a particularly unstable ligand binding-mode where the more frequent protein-ligand interactions occur with residues located on ECL2 (F168[45.52]) and TM6 (N253[6.55]), rather than TM7, which results in low H-bond occupancy with S277[7.42] in particular (S11 Fig).

As an extra control, additional MD simulations were performed in duplicate in a POPC membrane with or without bound adenosine (summarized in S12 Fig). Like in DOPC,

adenosine displays a similar level of positional instability when bound to the receptor in POPC (S12 Fig). Nevertheless, in both replicas the receptor is observed to reach an intermediate conformation similar to that observed in DOPC with bound adenosine (S12 Fig). This can be contrasted with the apo state in POPC where the receptor is observed to remain in an inactive conformation in both replicas and ECL2 is observed to move towards the empty orthosteric pocket by up to 2.0 Å (S12 Fig) in a similar manner to that observed with the apo state in DOPC. This suggests that DOPC and POPC membranes have similar effect (or lack of it) on A2aR with or without bound agonist, as might be expected.

## Apo A2aR fluctuates between inactive and intermediate states in DOPG membrane

Over quadruplicate MD simulations with apo A2aR in a DOPG membrane, ECL2 is broadly maintained in its initial crystal conformation (average final RMSD of 2.8 Å (0.9 SD), Fig 5) in similar fashion to the adenosine-bound receptor in DOPC (S4 Fig). This is different to that of apo A2aR in DOPC where ECL2 is observed to change conformation by approaching the open orthosteric pocket (Fig 3). This suggests that the extracellular side of the receptor behaves differently in an anionic membrane environment, which is related to conformational changes in the TM domain that contribute to conformational restriction of ECL2. In particular, this involves inward movement of TM7 (see below) that narrows the orthosteric pocket, which has previously been described in agonist-bound A2aR crystal structures [27]. In more detail, within 500 ns, similar to the behaviour of A2aR with bound adenosine, there is an initial change of conformation in TM3, TM5, TM6, and TM7. In particular, in two of four replicas (#1 and #2), TM3 shows noticeable upward movement by an average of 2.1 Å (0.3 SD) (Fig 5G), which is most sustained in replica #1. Likewise, in three of four replicas (#1, #2, #4), TM5 (S9 Fig) and TM6 move outward, while TM7 shifts inwards (Fig 5D and 5E) as their respective distances from TM3 change (mean (SD): 11.2 Å (1.1) and 12.1 Å (0.9), respectively). These respective outward and inward helical movements are synchronized, occurring at 250 ns in replicas #1 and #4, and after 1.4 μs in replica #2. However, it is noticeable that TM7 movement occurs shortly after those in TM6 take place first (Fig 5D and 5E). Therefore, without bound ligand acting on TM7, TM6 conformational changes need to be triggered first by protein-lipid intracellular interactions, such as those provided by anionic DOPG phospholipids. As has been previously shown with homologous β2-adrenergic and CB1 receptors, protein-DOPG H-bonds and electrostatic interactions are able to exert an outward pull on TM6 [50, 53]. These initial intracellular conformational changes in TM6 leave a space next to TM3, which TM7 can occupy by spontaneously moving inwards, creating intracellular and extracellular conformational changes that affect other receptor regions such as the G protein binding-site or ECL2.

In similar fashion to β2-adrenergic and CB1 [50, 53], DOPG is able to make specific protein-lipid interactions with A2aR through its negatively charged phosphate or polar head groups with positively charged or polar residues on the intracellular side of the receptor (Fig 6). As a result, time-averaged membrane thickness profiles across DOPG-containing MD simulations reveal that DOPG phospholipids cluster more strongly around A2aR than DOPC (Fig 6A–6C). In particular, four high occupancy DOPG interaction sites at specific intracellular receptor regions can be identified (Fig 6B and 6C). These correspond to areas between TM1 and H8, between TM3-TM4 and intracellular loop 2 (ICL2), and on the respective intracellular surfaces of TM5 and TM6 (Fig 6D–6G). In these sites, specific protein-lipid H-bonds and electrostatic interactions can be identified, including DOPG interaction with: i) a pair of tryptophan residues at the intracellular side of TM1 (W29$^{1.55}$ and W32$^{1.58}$), ii) neighbouring arginines on TM4 and ICL2 (R111$^{34.52}$, R120$^{4.41}$), iii) a pair of arginines on TM5 (R199$^{5.60}$,

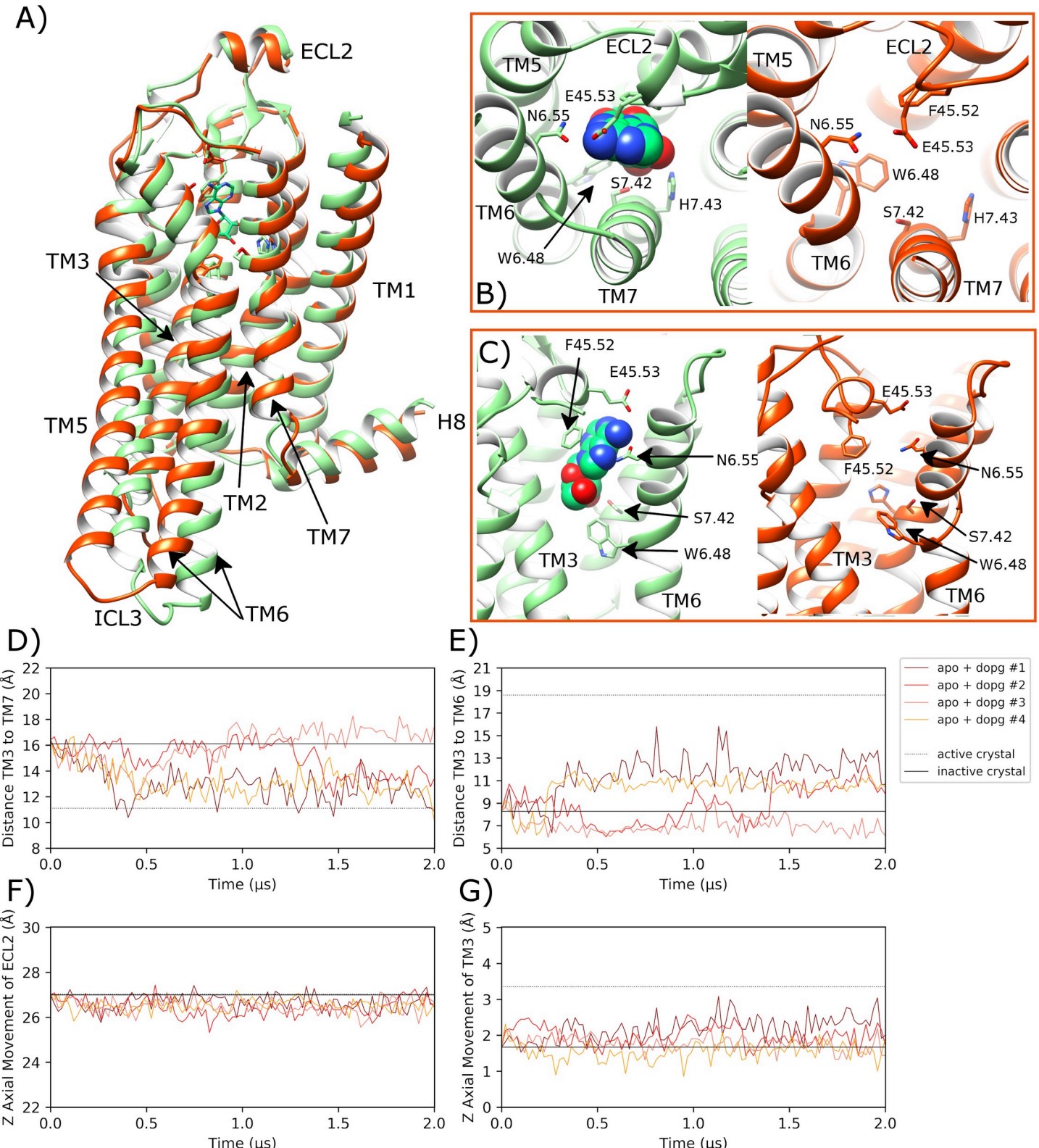

**Fig 5. Transition towards an alternative intermediate conformation of apo A2aR in a DOPG membrane.** A) Comparison of the intermediate state crystal structure of A2aR (PDB entry: 2YDO, light green) with bound adenosine and an MD-generated apo conformation achieved within a DOPG membrane (orange, belonging to replica #1 from 1.6 μs), showing B) and C) selected residues delineating the orthosteric pocket. ECL2 and TM helices are labelled where applicable. D) Distance between

TM3-TM7 (from Cα atoms of R102$^{3.50}$ and Y288$^{7.53}$, respectively) during MD simulations, starting from the inactive crystal structure (PDB entry: 4EIY). E) Distance between TM3-TM6 (from Cα atoms of R102$^{3.50}$ and E228$^{6.30}$, respectively). F) and G) Vertical movement of ECL2 and TM3, respectively. MD simulations are performed in quadruplicate. Corresponding flat-lines show the observed distance in the active (PDB entry: 6GDG) and inactive (PDB entry: 4EIY) A2aR crystal structures.

R206$^{5.67}$), iv) a lysine-histidine couple on TM6 (H230$^{6.32}$, K233$^{6.35}$). Each of these protein-lipid interactions is observed across all replicas in a DOPG membrane and marked by particularly stable lipid binding modes. For example, in several trajectories, a single DOPG molecule closely interacts with residue H230$^{6.32}$ continuously for 2 μs or, in others, one lipid is exchanged for another after 500–1000 ns and the interaction is re-established (S13 Fig).

By way of comparison, only two high occupancy protein-lipid interaction sites are observed in DOPC membranes: between TM1-TM2 and TM4 and, to a lesser extent, between TM3 and TM4 on the other side of the receptor (Fig 6A). It is particularly noticeable that DOPC phospholipids are not seen to cluster around TM5 or TM6 like with DOPG (Fig 6B and 6C). Interestingly, the TM1-TM2-TM4 external cavity is recognized as containing the cholesterol consensus motif or CCM, which in active A2aR appears to favourably bind membrane cholesterol and could account for its experimentally observed enhancement of receptor signalling [69, 77]. Interestingly, in our MD simulations, this cavity instead sequesters DOPC phospholipids, which might provide an opposite effect to cholesterol, potentially suggesting competition between these two molecules. In terms of comparison between DOPG and cholesterol, the binding of phospholipids to the intracellular surfaces of TM5 and TM6 may potentially share similarities with a reported second binding-site of cholesterol in a hydrophobic external crevice between TM5 and TM6 [104]. As cholesterol is an uncharged molecule, its precise mode of interaction is likely different to the electrostatic nature of anionic DOPG, meaning these two molecular species might theoretically be able to co-bind the receptor in this area. However, the exact function of cholesterol binding in this location in terms of receptor activation is difficult to verify as it was observed within MD simulations of the antagonist-bound inactive receptor (which remains inactive) and because crystal structures of active A2aR lack co-crystallized cholesterol [104].

Of the three replicas in a DOPG membrane where TM6 conformational change is observed (#1, #2, #4), concomitant rotameric changes in the sidechain of W246$^{6.48}$ are seen as it moves from g+ to trans (S8 Fig). This feature is similar to that observed with bound adenosine in a DOPC membrane and suggests that W246$^{6.48}$ acts as an activation switch (or at least facilitator to an intermediate receptor state) even if no agonist is bound. Also noteworthy, different degrees of TM6 conformational change appear to correlate with the duration W246$^{6.48}$ maintains a continuous trans conformation. For example, in replicas #2 and #4, where W246$^{6.48}$ adopts a trans conformation for ~250 ns and ~800 ns, respectively, the distance between ionic lock residues located on TM3 and TM6 (R102$^{3.50}$ and E228$^{6.30}$) increases up to a maximum of 12.4 Å and 11.9 Å, respectively, as TM6 moves outwards (Fig 5E). This level of conformational change in TM6 is comparable to that which occurs in a DOPC membrane when adenosine is bound and suggests formation of a similar intermediate receptor state (S14 Fig). However, in replica #1, W246$^{6.48}$ maintains a trans conformation for 1.5 μs after an initial rapid switch and TM6 is seen to undergo a more considerable conformational change, establishing a distance from TM3 of up to 15.8 Å (Fig 5E). Although only transient in nature, this degree of TM6 conformational change is close to that observed in the active G protein-bound A2aR crystal structure [25]. This suggests formation of an intermediate receptor conformation that is "enhanced" and different from the others (Fig 5A). Indeed, this is further supported by TM6 RMSD, which increases up to 6.0 Å (S6 Fig) while RMSD of the TM domain temporarily reaches 2.2 Å compared to the active crystal state before regressing (S5 Fig). Together these

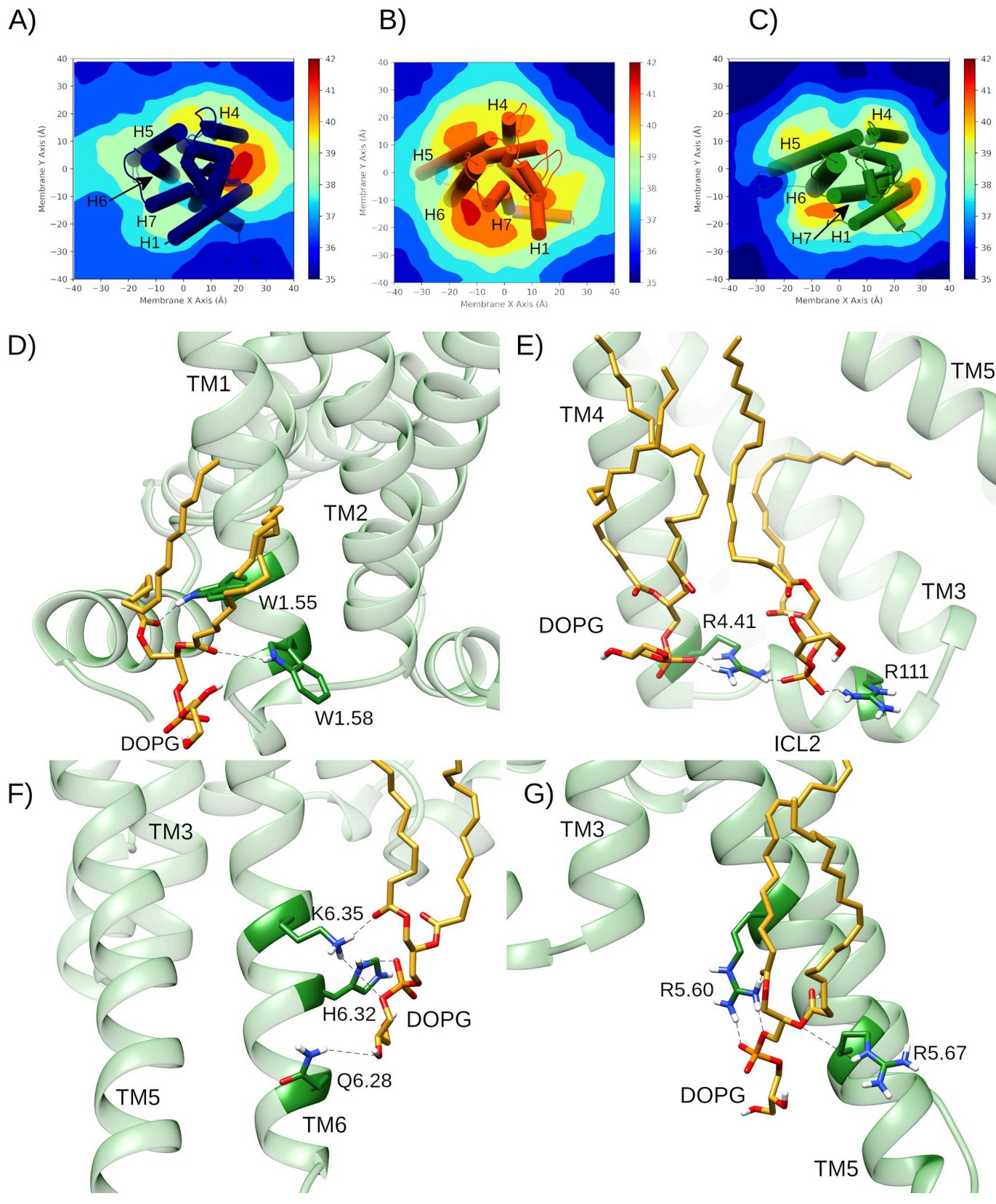

**Fig 6. Membrane thickness and allosteric protein-lipid interactions during MD simulations of A2aR.** Average membrane thickness measurements across 2 μs MD simulations of A2aR in A) DOPC with bound adenosine (replica #1), B) DOPG in apo (replica #2), and C) DOPG with bound adenosine (replica #1). D-G) Specific intracellular protein-lipid interactions of adenosine-bound A2aR in a DOPG membrane during 2.0 μs MD simulation (replica #1). D) Allosteric interaction between polar residues on TM1 (W29$^{1.55}$ and W32$^{1.58}$) in A2aR (green) with a DOPG lipid (gold). Electrostatic and H-bond interactions between charged/polar residues on: E) TM4 and ICL2 (R111$^{34.52}$, R120$^{4.41}$), F) TM6 (Q226$^{6.28}$, H230$^{6.32}$, K233$^{6.35}$) and G) TM5 (R199$^{5.60}$, R206$^{5.67}$) with DOPG lipids (gold).

data suggest the receptor reaches a transient conformation close to the active state before returning to a less-active, more conventional intermediate state. On the other hand, replica #3 shows that A2aR in a DOPG membrane can remain in an inactive-like state throughout. This demonstrates that the physical effect of a DOPG membrane on apo A2aR is variable over a two-microsecond time-period.

Finally, in addition to allosteric protein-lipid interactions formed at the receptor surface, the outward orientation of TM6 in a DOPG membrane can be further assisted by formation of a specific protein-lipid salt-bridge at the intracellular side of the receptor interior, which has previously been observed to occur in homologous β2-adrenergic and CB1 receptors [50, 53, 105]. This particular allosteric interaction is possible because of the internalization of a single DOPG lipid from the lower leaflet into the G protein binding-site between TM6 and TM7 (Fig 7A). Specifically, it is stabilized by an electrostatic interaction between DOPG phosphate group and R102$^{3.50}$ sidechain on TM3 (Fig 7A), which normally forms part of the TM3-TM6 ionic-lock in the inactive receptor state. This protein-lipid interaction is found to occur on a consistent basis in all four replicas in DOPG and is stable in three, persisting for more than 1.5 μs in each case (Fig 7C). Furthermore, it appears to be especially suited to DOPG lipids as it is not observed in any of the simulations performed in DOPC. This is likely due to the bulky hydrophobic head-group of phosphatidylcholine (PC) lipids, which are seemingly unable to penetrate the TM6-TM7 gap to the same extent as the smaller hydrophilic head-group of phosphatidylglycerol (PG) lipids [105]. However, despite the prominence and eye-catching nature of this allosteric interaction, it does not by itself guarantee receptor activation or transition to an intermediate receptor conformation as demonstrated by lack of receptor activation in replica #3 (Fig 5). In this particular replica, allosteric interaction between a DOPG lipid and lysine-histidine couple on the exterior of TM6 (H230$^{6.32}$, K233$^{6.35}$) is less stable compared to other replicas. Therefore, lipid interaction with TM6 appears more important than penetration of a lipid into the G protein binding-site between TM6 and TM7.

## Adenosine-bound A2aR in DOPG membrane reaches active conformation

In MD simulations of adenosine-bound A2aR in DOPG, unlike in DOPC, the agonist displays higher positional stability across all four replicas and remains in a binding pose that is consistent with the adenosine-bound crystal [93], as well as the NECA-bound active crystal state [25] (Fig 8B). This is confirmed by an average ligand RMSD of 1.4 Å (0.6 SD) (Fig 8D) and average ligand conformational fluctuation (RMSF) of 1.1 Å (0.6 SD) (Fig 8E) across all four replicas. As a result, bound adenosine sustains key protein-ligand interactions for longer compared to corresponding simulations in a DOPC membrane. For example, regarding ECL2, aromatic stacking between F168$^{45.52}$ and the bicyclic ligand core is observed for 55.2% (11.9 SD) of the time on average, while H-bonding with E169$^{45.53}$ shows an average occupancy of 29.2% (9.5 SD) (S15 Fig). These interactions appear to consistently shift ECL2 upwards by an average of 1.2 Å (0.4 SD) (S4 Fig) with an average loop RMSD of 3.1 Å (1.1 SD) across all replicas. This conformational change is not observed as consistently with bound adenosine in a DOPC membrane (S4 Fig), which suggests protein-DOPG allosteric interactions also have an effect in agonist-bound receptor states. Furthermore, in a DOPG membrane, the ribose hydroxyl

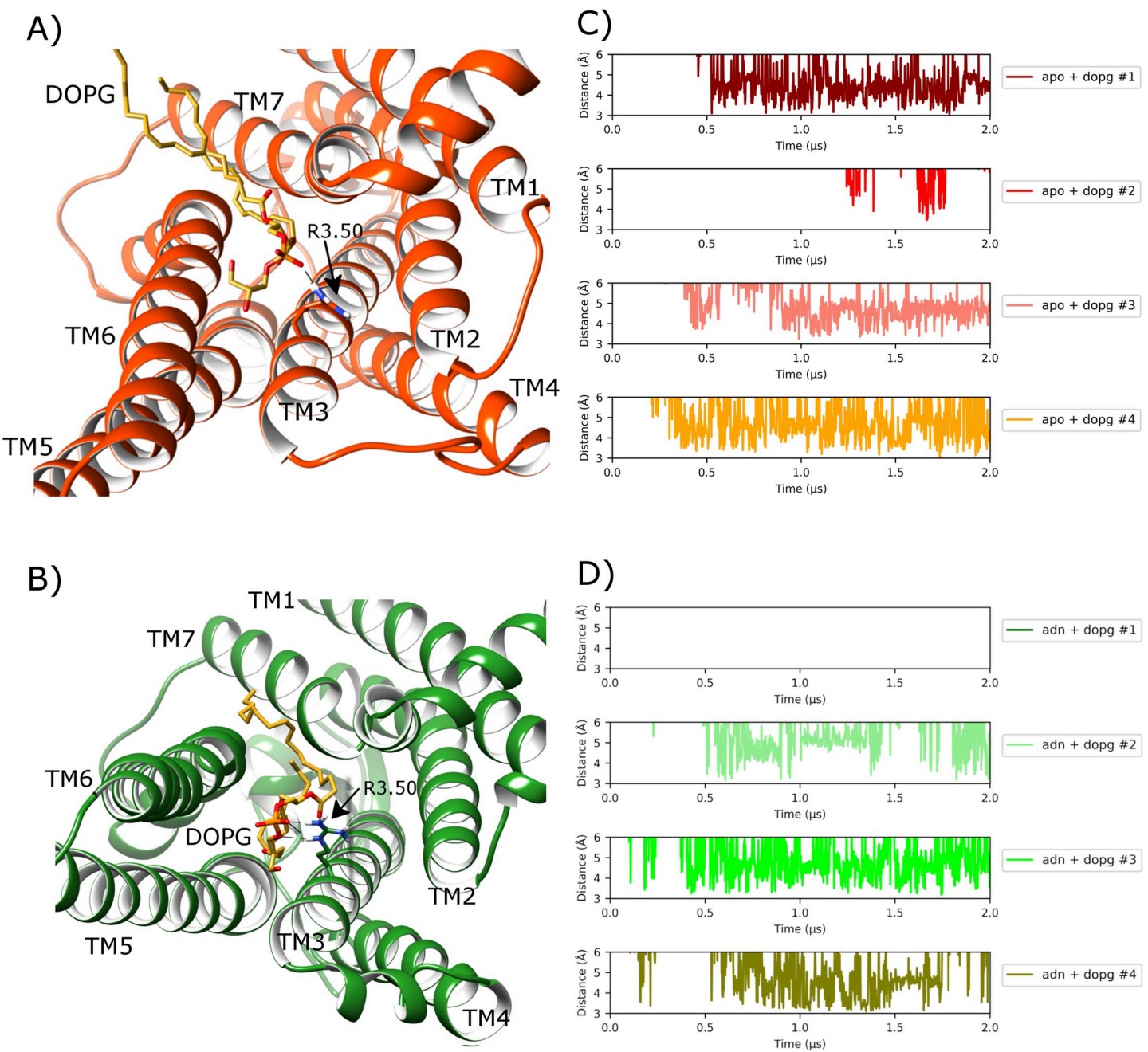

**Fig 7. Protein-lipid allosteric interaction with the ionic-lock in MD simulations of A2aR in a DOPG membrane.** A) Intracellular view of A2aR where ionic-lock residue R102[3.50] electrostatically interacts with a DOPG lipid, which intrudes between TM6 and TM7 into the G protein binding-site in apo state (orange) (belonging to replica #4 from 1.8 µs), and B) with bound adenosine (green) (belonging to replica #2 from 1.6 µs). Protein-lipid interaction distance over time between R102[3.50] sidechain and lipid phosphate group in four replicas of A2aR in DOPG membrane in C) apo state and D) adenosine-bound (ADN), respectively.

groups of adenosine make stronger H-bonds with S277[7.52] and H278[7.53] on TM7 with average occupancies of 59.3% (13.8 SD) and 41.8% (19.4 SD), respectively (S15 Fig). At the same time, N253[6.55] forms a stable protein-ligand H-bond with average occupancy of 95.0% (5.4 SD) across all replicas in DOPG (S15 Fig). In light of these effects on ligand stability and extracellular conformational changes, adenosine-bound A2aR displays the same four protein-DOPG

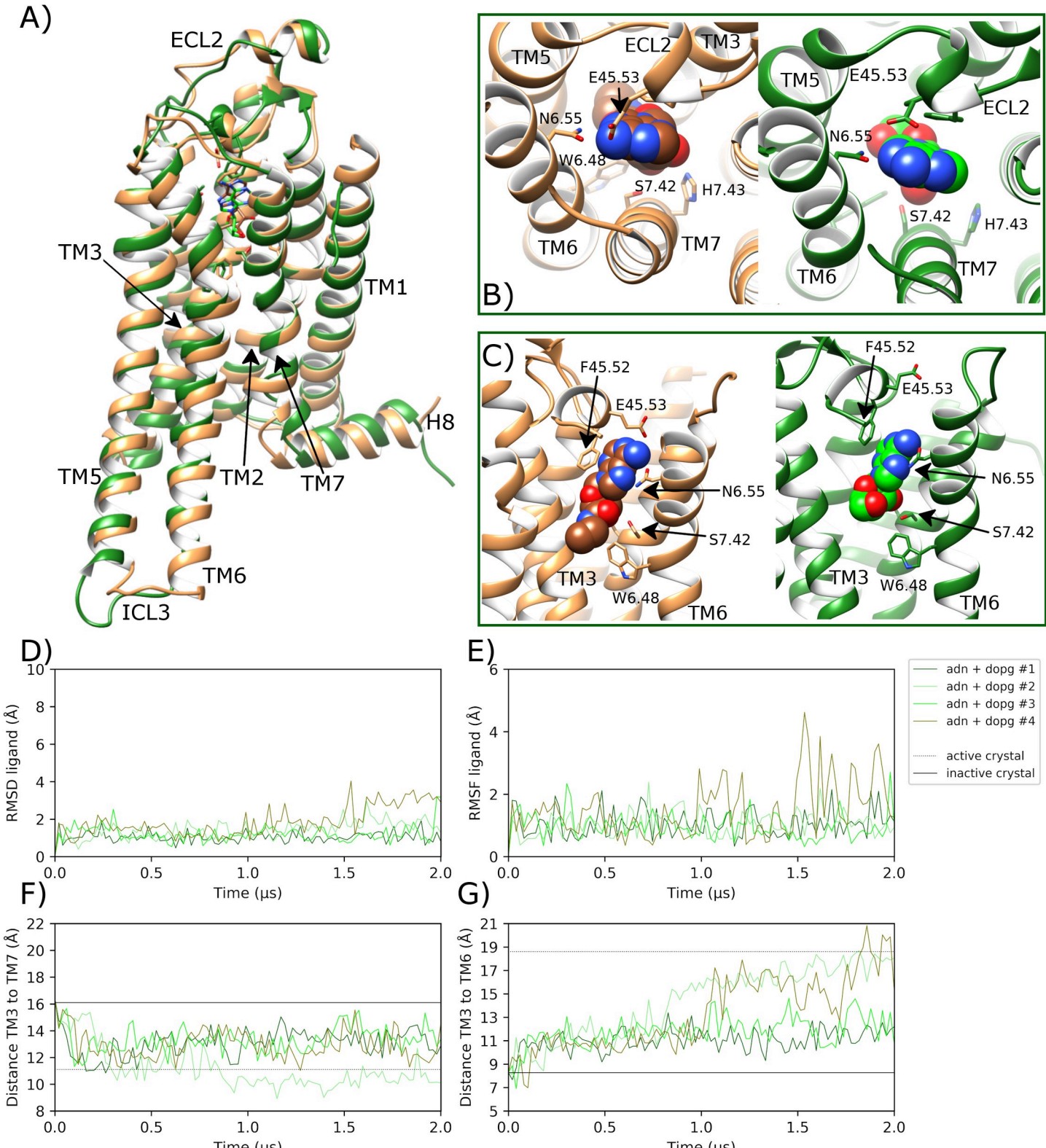

**Fig 8. Transition towards an active-like state of A2aR in MD simulations with bound adenosine in a DOPG membrane.** A) Comparison of the MD-generated conformation of A2aR bound to adenosine (ADN) within a DOPG membrane (green, belonging to replica #2 from 0.7 μs) with the active crystal structure of A2aR with bound NECA (brown, PDB entry: 6GDG), showing B) and C) protein-agonist interactions in the orthosteric pocket with adenosine and NECA atoms displayed as

spheres. ECL2 and TM helices labelled where applicable. D) RMSD of bound adenosine (calculated with respect to initial docking pose). E) Conformational fluctuation (RMSF) of adenosine. F) Distance between TM3-TM7 (from Cα atoms of R102$^{3.50}$ and Y288$^{7.53}$, respectively) during MD simulations starting from the inactive crystal structure (PDB entry: 4EIY). G) Distance between TM3-TM6 (from Cα atoms of R102$^{3.50}$ and E228$^{6.30}$, respectively). MD simulations are performed in quadruplicate. Corresponding flat-lines show the observed distance in the active (PDB entry: 6GDG) and inactive (PDB entry: 4EIY) A2aR crystal structures.

allosteric interaction hot-spots as seen with the apo state in DOPG: TM1-H8, TM3-4-ICL2, TM5, and TM6 (Fig 6B and 6C). Closer inspection reveals the same specific intracellular protein-lipid interactions as observed in the apo state (Fig 6D–6G) but with increased stability of interaction between the lysine-histidine couple on the intracellular surface of TM6 (H230$^{6.32}$, K233$^{6.35}$) and a DOPG molecule (S13 Fig). This is suggestive of an allosteric pathway acting through TM6 for two-way communication between bound agonist in the orthosteric pocket and bound intracellular lipids.

As a consequence of protein-lipid electrostatic interactions at the intracellular side of the receptor (Fig 6D–6G) and stable adenosine binding, profound receptor conformational changes are observed in TM helices. Firstly, in replicas #1 and #3, TM3 moves upwards by a maximum of 1.9 and 1.6 Å, respectively, and by an average of 0.8 Å (0.4 SD) across all replicas (S7 Fig). This vertical shift matches that observed in the fully active crystal structure of A2aR and results in an average L95$^{3.43}$ RMSD from the inactive state of 3.1 Å (0.8 SD) (S7 Fig), as TM3 undergoes axial rotation. Likewise, across all replicas, respective TM3-TM5 (S9 Fig) and TM3-TM7 (Fig 8F) distances rapidly reach and even surpass those seen in the fully active crystal structure (mean (SD) distances: 22.1 Å (2.6) and 12.5 Å (1.5), respectively). This suggests the receptor is activating across all replicas with inward movement of TM7, upwards movement of TM3 and outward movement of TM5 all occurring within 100 ns.

Meanwhile, the sidechain of W246$^{6.48}$ rapidly moves from g+ to trans within 200 ns in three out of four replicas (S8 Fig), which assists more consistent conformational change in TM6 (Fig 8G). However, only in replicas #2 and #4, does TM6 continue to move outwards enough to establish a TM3-TM6 distance that matches the active crystal state (maximum values of 18.6 Å and 20.4 Å, respectively) within the final 400 ns of each replica (Fig 8G). This degree of conformational change corresponds to a maximum TM6 RMSD of 7.8 Å and 8.4 Å, and a minimum of 2.3 Å compared to TM6 in the active crystal (S6 Fig). This degree of movement can be directly correlated with particularly stable DOPG binding to the intracellular surface of TM6 via positively charged residues: H230$^{6.32}$ and K233$^{6.35}$ (S13 Fig). Likewise, following a similar trend to TM6, receptor RMSD exceeds 4.9 Å in replicas #2 and #4, and reaches a minimum of 2.2 and 2.3 Å compared to the active crystal (S5 Fig), respectively. These conformational changes allow water molecules to penetrate the core of the receptor, forming a continuous channel from the bottom of the orthosteric pocket to the intracellular space, unlike in the apo state in DOPC where the core is mostly desolvated (Fig 9A and 9B). In addition, the adenosine-bound state shows less water density in its orthosteric pocket compared to the apo state, which is fully hydrated. This is because the agonist displaces several water molecules as well as creating a narrower binding pocket, which tends to eliminate even more waters, as has been previously proposed in GPCR activation [27]. Interestingly, the observed movement of W246$^{6.48}$ disrupts its water-mediated interaction with D52$^{2.50}$ seen in the apo state in DOPC (Fig 9C and 9D). Indeed, recent NMR studies have suggested that W246$^{6.48}$ is involved in the same receptor micro-switch as D52$^{2.50}$ [63]. This could mean that in order for TM6 to move outwards and for TM3 to move upwards, the water network that connects W246$^{6.48}$ and D52$^{2.50}$ needs to be broken. As such, when W246$^{6.48}$ toggles into its trans position, it makes a new water-mediated H-bond with bound adenosine via a pair of water molecules (Fig 9D). This exact interaction is not observed in the adenosine-bound

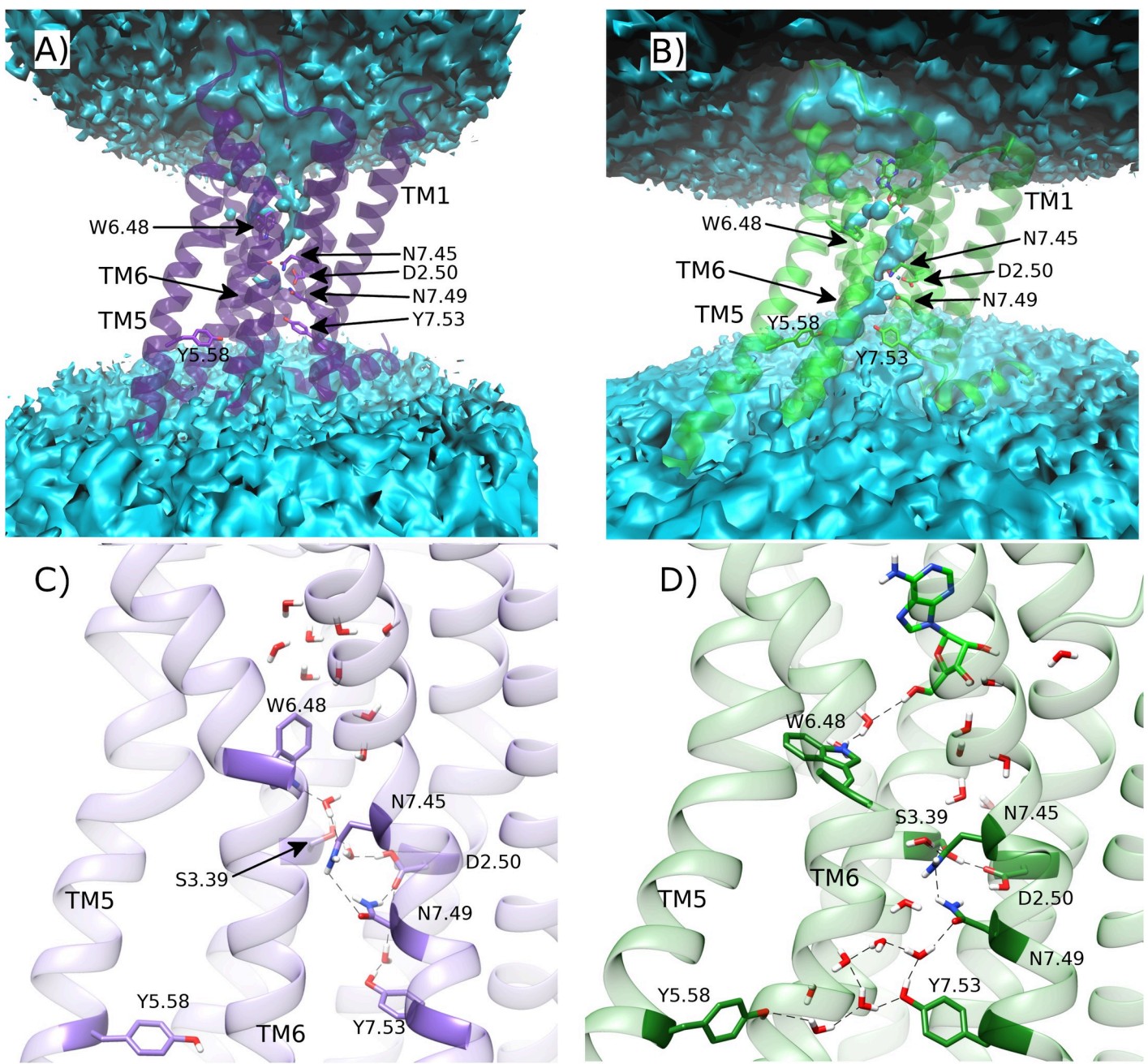

**Fig 9. Comparison of water-mediated polar networks in the core of A2aR during MD simulations.** Average water density of A) apo A2aR in DOPC membrane, and B) A2aR bound to adenosine within a DOPG membrane (green) over respective replica #1 trajectories. Respective water molecule distribution snapshots at C) 1.5 μs and D) 1.3 μs. Water-mediated hydrogen bonds are represented as dotted lines.

intermediate A2aR crystal structure even though the same pair of water molecules are co-crystallized (S16 Fig). Nevertheless, our MD simulations suggest it may play a role in the transition to a fully active receptor state. In addition, a direct interaction is formed between D52$^{2.50}$ and S91$^{3.39}$ in the core of the receptor, which can be observed in the adenosine-bound crystal structure (S16 Fig). These residues are connected to N280$^{7.45}$, N284$^{7.49}$ and Y288$^{7.53}$ on the NPxxY motif of TM7, as well as Y197$^{5.58}$ on TM5, via numerous water-mediated interactions (Fig

9D). Perhaps most notably, waters also play a role in the reorientation of Y197[5.58] and Y288[7.53] side-chains towards each other, which form a stabilizing residue-residue interaction in G protein-bound active crystal structures [24, 25, 106]. Interestingly, some of these waters have also been co-crystallized in the high-resolution inactive and medium-resolution adenosine-bound A2aR crystal structures [29, 93] (S16 Fig), supporting their involvement in A2aR activation as observed in our MD simulations and in other GPCRs [33].

It should be noted however that, although A2aR obtains a full set of active-like conformational features in replicas #2 and #4 (Fig 8), no one single receptor conformation is consistently stabilized. This is likely due to the absence of a bound G protein, which if present would presumably stabilize the precise receptor conformation observed in the G protein-bound A2aR crystal [25]. Instead, in our system, the activated free receptor appears to explore a wider landscape of active-like metastable conformations, especially in replica #4 where TM5 and TM6 obtain orientations more outward than the active crystal structure (S17 Fig). Indeed, recent NMR data has identified two different and distinct active conformational states of A2aR [61]. This appears to fit with our data as we are able to identify at least two distinct active-like receptor conformations, which have appreciable difference in TM5, TM6 and TM7 (S18 Fig). Conformational fluctuation observed in active-like states of A2aR may also be a consequence of allosteric intrusion of a single DOPG lipid between TM6 and TM7 into the G protein binding-site of the receptor (Fig 7). This protein-lipid allosteric interaction was previously observed with the apo state in DOPG and reoccurs here in replicas #1, #3 and #4, although not in replica #2. Interestingly, this seems to exert a destabilizing effect on the active-like receptor conformation obtained in replica #4, which sees TM6 moves further outwards in the process. On the contrary, in replica #2, with no internalized lipid molecule, receptor conformational change appears more gradual and stable. Therefore, this particular protein-lipid interaction may lead to alternative active-like receptor conformations or may simply be detrimental to sustained receptor activation in general. On the contrary, in replica #1 and 3, TM6 does not reach the fully active conformation and instead the receptor remains in an intermediate-like state, which is mostly consistent with the adenosine-bound A2aR crystal structure [93]. This demonstrates that even in conducive conditions, with bound adenosine and anionic phospholipid membrane, full receptor activation is not guaranteed within a time period of two microseconds.

## NECA is a more potent agonist than adenosine

In order to contrast the endogenous agonist (adenosine) with the effects of a potent synthetic agonist, quadruplicated MD simulations were performed from the inactive state with bound NECA in DOPC and DOPG membranes. These MD simulations reveal several similarities but key differences between the biological effect of the two agonists. Firstly, bound NECA is comparatively more stable in the orthosteric pocket in both membrane types than adenosine (S19 Fig, S20 Fig). However, NECA stability is enhanced by a DOPG membrane (S20 Fig), like that observed with adenosine. In particular, in a DOPG membrane relative to DOPC, NECA makes a closer and more stable H-bond with residue S207[7.42] on TM7 (mean (SD) H-bond occupancies of 67.8% (18.8) and 55.2% (16.3), respectively; S21 Fig, S22 Fig). Likewise, when comparing both agonists in a DOPC membrane, NECA makes a stronger and more stable H-bond with S207[7.42] than adenosine (mean (SD) occupancies of 55.2% (16.3) and 30.4% (23.6), respectively; S11 Fig, S21 Fig). This difference may explain the extra potency of NECA, which in DOPC results in the receptor reaching an intermediate conformation in 4/4 replicas, and in DOPG results in the receptor obtaining an active-like conformation in 3/4 replicas and at least partial activation in the fourth (S20 Fig, S21 Fig). This shows, like with bound adenosine, that a DOPG membrane enhances the effectiveness of NECA through positive allosteric

modulation of bound phospholipids on the intracellular side of the receptor, which interact in the same way as observed previously e.g. with residue R102$^{3.50}$ (S23 Fig). The cumulative effects of bound phospholipids and agonist are observed to have profound consequences on the overall population of receptor conformations, which can be defined according to key inter-helical TM3-TM6 and TM3-TM7 distances (Fig 10, S24 Fig, S25 Fig). Furthermore, MD simulations with the more potent NECA agonist clearly reveal that agonist-mediated receptor activation results in approximately equal formation of two distinct active-like receptor conformations, as previously proposed by NMR experiments [41]. One of these conformations has a comparatively wider separation between TM3, TM6, and TM7 helices and resembles the G protein-bound active crystal structure, while the other has relatively closer TM helices but is still active-like (Fig 10).

## Protein-protein docking reproduces the G protein-bound A2aR crystal interaction

In order to validate that A2aR bound to adenosine or NECA in a DOPG membrane is indeed able to reach an active receptor conformation at a functional level, we docked the co-crystal-lized G$_s$α protein[25] into MD-generated active-like conformations of A2aR obtained during each replica (as well as re-docking back into the active A2aR crystal structure as a control). In the two replicas with bound adenosine or three replicas with bound NECA where A2aR reaches the fully active conformation, G$_s$α is able to dock into the intracellular G protein binding-site without any steric clashes with TM helices or ICL1 and 2 (Fig 11). Mainly, adequate docking is achieved due to proper separation of TM5/TM6 from TM3. The re-docking of the active crystal structure of A2aR with G$_s$α generates an interface docking score of -7.9 (no units, I_sc range from 0.0 to -10.0; more negative is better with -5.0 representing a threshold for respectable interaction [107]) (Fig 11A, S3 Table), while an MD-generated conformation with bound adenosine in DOPG from replica #2 with G$_s$α gives a score of -7.7 and an RMSD for G$_s$α of 0.8 Å (Fig 11B, S3 Table). Likewise, with bound NECA in DOPG, replica #3 with G$_s$α also gives a score of -7.7 and an RMSD for G$_s$α of 1.1 Å (S3 Table). This supports a proper mode of interaction between G$_s$α and A2aR bound to adenosine or NECA in a DOPG membrane. On the contrary, MD-generated receptor conformations in an intermediate or inactive state (in DOPG without bound agonist or in DOPC with or without bound agonist) only obtain I_sc docking scores from -3.3 to -6.1 and G$_s$α RMSD of 4.9–8.3 Å (S3 Table).

## Discussion

The relatively recent availability of high-resolution crystal and cryo-EM structures of GPCRs in different states has facilitated understanding of factors governing their process of activation. These structures have been informative in elucidating the molecular basis for A2aR activation and ligand binding in particular, and have also opened up avenues for studying the role of receptor dynamics and receptor conformation ensembles [42, 61–64]. Interestingly, A2aR has also been crystallized with different interacting lipids [29, 80, 81, 88, 92, 94], which highlights the importance of phospholipids in the process of stabilizing different receptor conformations [69, 77]. Here we have performed unbiased high-throughput MD simulations of A2aR, starting from the inactive crystal structure [29] with or without bound agonists: adenosine or NECA, in PC or PG homogeneous lipid environments to give a deeper understanding of the receptor activation process and cooperative forces between agonist and phospholipids.

From our results, when A2aR is embedded in different phospholipid membranes, anionic PG lipids preferentially cluster around four intracellular areas of the protein: TM1-H8, TM3-TM4-ICL2, TM5, and TM6. On the other hand, neutral PC lipids cluster around just

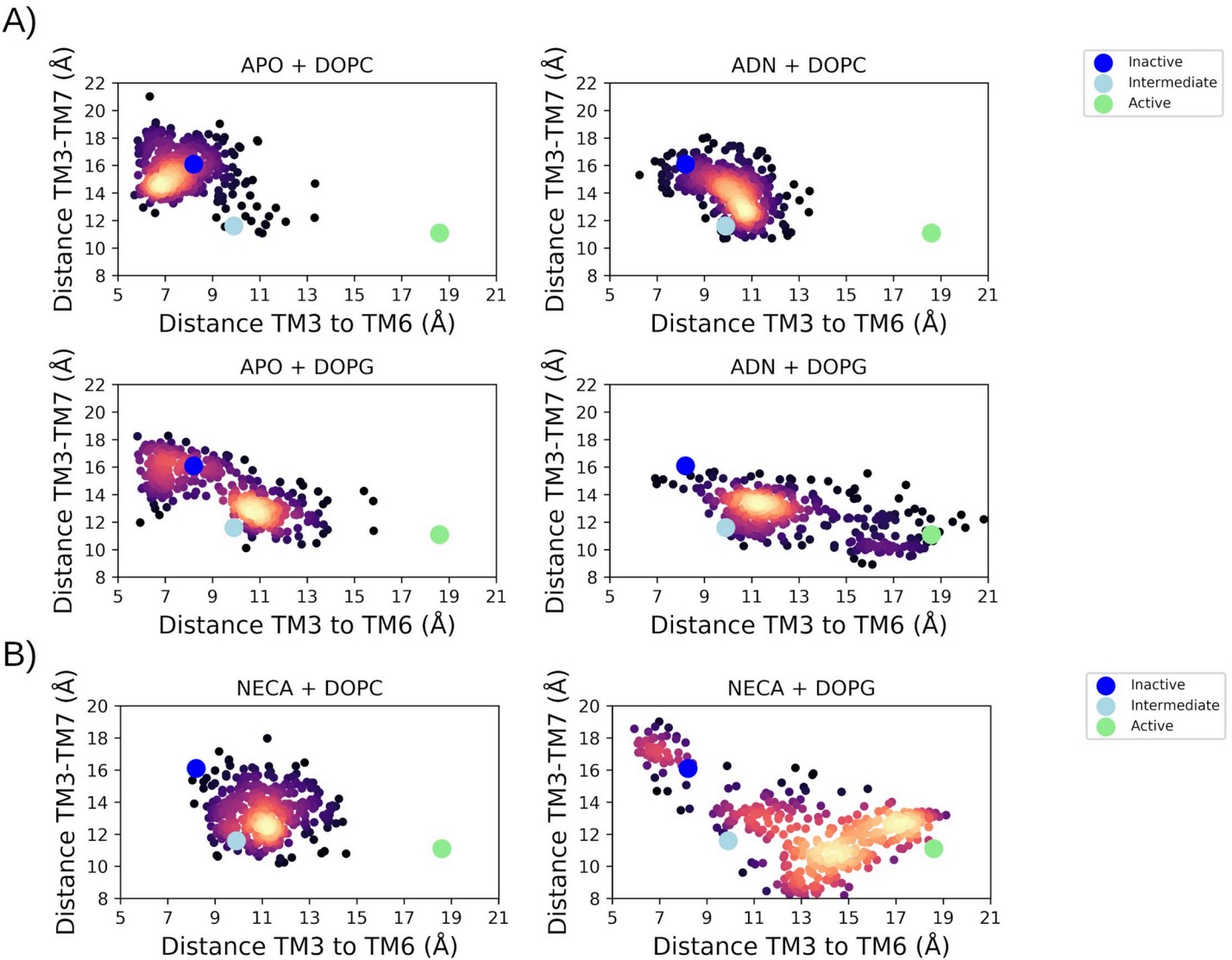

**Fig 10. The two-dimensional collective variable (CV) space of A2aR activation and receptor conformations during respective MD simulations: A) apo or with bound adenosine (ADN); B) with bound NECA**. CVs correspond to distances between ionic lock residues R3.50 and E6.30 (TM3-TM6), and residues R3.50 and Y7.53 (TM3-TM7). The reference distances from inactive, intermediate and active crystal structures are shown as blue, cyan, and green dots, respectively. Frequency of receptor conformation is represented by a "heat" scale (low: black, high: yellow).

two intracellular areas: a cavity formed by TM1-TM2-TM4 and to a lesser extent TM3-TM4. The primary interaction site for PC is consistent with a previously described intracellular interaction site for cholesterol, which is also a neutral and mostly hydrophobic molecule. Through its binding, cholesterol has been reported to stabilize the active receptor conformation [69, 77]. This poses the possibility of PC lipids and cholesterol competing at the same location and exerting different influences over receptor activity. Likewise, cholesterol has been suggested to also bind between TM5 and TM6, which could conceivably potentiate allosteric effects of PG lipids in this area. Although an intriguing question, cholesterol binding is beyond the scope of this study, which as a relatively weak and slow process (compared to phospholipids) would necessitate much longer MD simulations and preclude execution of quadruplicates [108]. In terms of protein-phospholipid interactions on the extracellular side of the receptor, we also

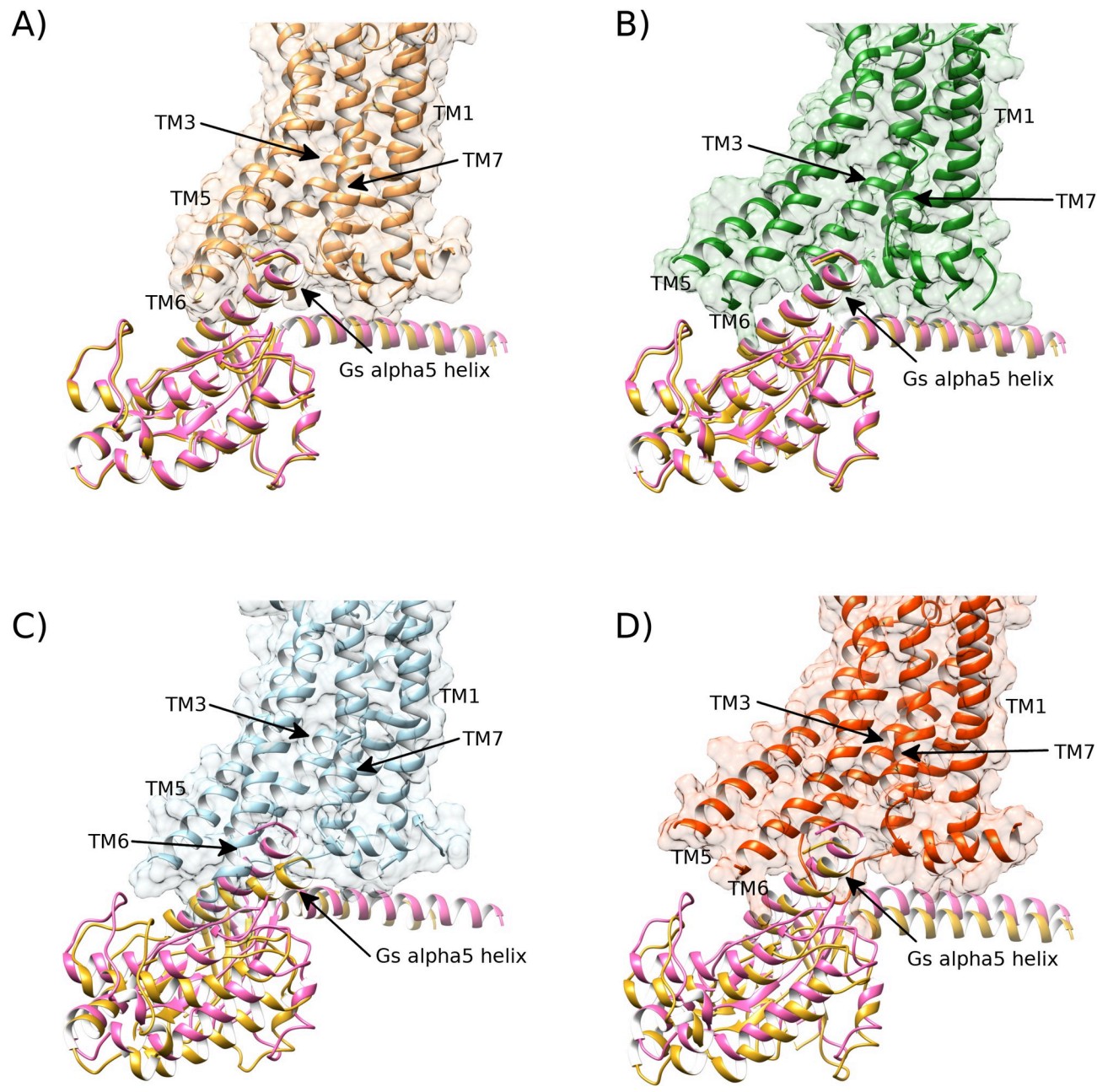

**Fig 11. The MD-generated receptor conformation of A2aR bound to adenosine in DOPG membrane is able to bind co-crystallized G$_s$-alpha protein in the same way as the active-state crystal structure (PDB id: 6GDG).** A) The crystal structure of the active state of A2aR (brown) bound to its co-crystallized G$_s$-alpha protein (pink, PDB id: 6GDG) and its re-docked G$_s$-alpha subunit superimposed (gold). B) MD-generated active-like conformation of A2aR bound to adenosine in a DOPG membrane (green, belonging to replica #2 from 1.6 µs) docks G$_s$-alpha protein (gold) in similar fashion to the active crystal (pink). C) Intermediate conformation of A2aR, bound to adenosine in DOPC membrane (blue, belonging to replica #4 from 1.3 µs) fails to properly dock G$_s$-alpha protein (gold) compared to its active crystal position (pink). D) Intermediate conformation of apo A2aR in DOPG membrane (red, belonging to replica #1 from 1.6 µs) partially docks G$_s$-alpha protein (gold) compared to its active crystal position (pink).

observe some binding of phospholipids in the hydrophobic interface between TM1 and TM7 as has been previously proposed in some GPCRs [109]. However, in our MD simulations these are not as physically stable as interactions at the intracellular side (hence, relative lack of clustering intensity in Fig 6) and are seemingly not specific to either PC or PG lipids. One potential

functional outcome of having several high occupancy intracellular allosteric sites for anionic PG lipids means that they may regulate multiple local conformational dynamics of the receptor. In particular, three of four PG-interaction hot spots correlate with positions of TM3, TM5 and TM6, which are key helices in the activation process of class A GPCRs, including A2aR [16, 27]. These specific protein-lipid interactions are mainly mediated by electrostatic contacts between positively charged sidechains and negatively charged lipid phosphate groups and tend to favour the outward movement of the cytoplasmic halves of TM5 and TM6, thus aiding intracellular conformational change. As such, there is a correlation between sustained binding of PG lipids to intracellular regions of TM6 and enhanced TM6 conformational change, resulting in more efficient G protein docking, rearrangement of the internal water-mediated polar network, tryptophan sidechain toggling, more stable protein-agonist H-bonding, and increased stabilization of ECL2. This creates a multidirectional allosteric network where different entities affect the action of others. ECL2 is thought to play an important role in receptor signalling, controlling access to the orthosteric pocket and recognition of agonists [110]. Indeed, disruption of the conserved disulphide bridge between ECL2 and TM3 largely diminishes receptor function [110, 111]. In DOPG, ECL2 tends to be stabilized in a higher position above the orthosteric pocket than in PC membranes (S26 Fig), which appears to assist TM3 conformational change and vice versa, as well as enhancing the stability of bound agonist. This indirect relationship between membrane phospholipids and ECL2 may provide a structural explanation for how GPCRs exhibit different ligand efficacies in different cell lines [110, 112]. On the contrary, PC lipids, which have a more bulky and hydrophobic head group than DOPG, are less suited for forming electrostatic or H-bond protein-lipid interactions with the receptor [50]. Thus, this absence of sustained protein-lipid interaction triggers fewer intracellular conformational changes in the receptor, preferentially favouring an inactive conformation, which, in turn, destabilizes agonist binding and promotes ECL2 flexibility.

As a result, the apo state of A2aR in PC membranes remains inactive, which prevents adequate G protein docking. This is expected and in agreement with previously published MD simulations [65, 72]. In contrast, two distinct intermediate receptor conformations can be identified in the apo state in a DOPG membrane, which depend partly on the stability of key allosteric protein-lipid interactions. One of these conformations is consistent with the intermediate state observed with bound agonist in PC membranes, as well as the agonist-bound A2aR crystal structure [93], whilst the other is "enhanced" with a more outwards TM6 conformation that is closer to an active-like state, albeit not physically stable. This is a striking result, because A2aR has been reported to signal across cellular membranes in the absence of agonists [42, 113], which our MD simulations suggest may be dependent on phospholipid content. Moreover, these two different intermediate receptor conformations are notable because NMR experiments have identified two distinct intermediates of A2aR, which were resolved in apo and agonist-stabilized states, respectively [41]. Taken together, this demonstrates that like other homologous GPCRs [50, 53], A2aR can be positively modulated by PG lipids even without bound agonist. On the other hand, neutral PC lipids induce little or no conformational change over a microsecond time-window and instead depend on agonist binding for reaching an intermediate receptor state. Indeed, it may be the case that a millisecond time period is required, or concomitant G protein binding, for the receptor to reach a fully active state in this type of membrane. These findings also raise the classical debate on induction vs selection in receptor activation [114]. Our results are compatible with both views because if an agonist is able to induce similar receptor conformational changes as the receptor is able to achieve by itself (in an appropriate membrane environment) then it would also be able to stabilize (select) an active receptor conformation if the orthosteric pocket is not obstructed [114–116].

Interestingly, the possibility that A2aR activation includes both induced fit and conformational selection mechanisms has been proposed by others on the basis of NMR experiments [61].

In agreement with previously published data [70], adenosine displays high mobility in the pocket of A2aR when embedded in a neutral PC membrane, while NECA shows noticeably greater stability. This translates into more consistent receptor activation by NECA, which is a more potent agonist. However, when adenosine is reasonably stable, the receptor transitions towards an intermediate conformation with TM3 moving partially upwards due to steric influences on this helix, as well as protein-ligand interactions with ECL2, which connects with TM3 via a disulphide bridge [110, 111]. This is interesting because such TM3 transitions occur less consistently in the apo state in DOPG despite favourable intracellular protein-lipid allosteric interactions. This suggests that agonist binding drives conformational change that favours a particular pathway of receptor activation, which leads to a more consistent intermediate conformation. Conversely, the pattern of receptor activation mediated by DOPG lipids directs the apo receptor to more diverse conformational changes, which potentially involve various intermediate conformations or different activation pathways, as has been recently proposed by MD simulations in a cholesterol-rich membrane [78]. Indeed, here in the apo state, the predominant activation pathway proceeds through TM6 first, while with bound agonist, a faster pathway proceeds through TM3 and TM7, and later TM6. Likewise, the existence of different receptor intermediate conformations suggest natural variation in the A2aR activation process, which could perhaps offer opportunities in drug design if a pathway through one intermediate conformation was deemed therapeutically more advantageous than another [106].

Agonist-bound A2aR in DOPG is able to activate via a two-step activation mechanism, starting from inactive to intermediate and then to an active-like state. Receptor activation begins with conformational changes in the orthosteric pocket and the triggering of microswitches in the receptor core followed by the inward shift of TM7, upwards axial tilt of TM3 and outward movement of TM5. This is combined with a partial outward displacement and rotation of TM6, which brings together the intracellular ends of TM5 and TM6, which gradually adjust their inter-helical contacts, consistent with the intermediate agonist-bound crystal structure [93]. As a result, this modifies water-mediated hydrogen-bond networks in the core of the receptor, ultimately leading to the formation of a continuous water column extending from the orthosteric pocket to the intracellular side of the receptor and connecting residues Y288$^{7.53}$ and Y197$^{5.58}$, which have previously been shown to be an activation-associated micro-switch [93, 106]. Secondly, TM6 moves even further out, achieving the fully outward conformation of the G protein-bound active crystal structure [24, 25]. This extra conformational change in TM6 appears to be facilitated by the membrane and not agonist, at least according to our observations in a microsecond timescale. As a consequence, G$_s$α protein can be docked into agonist-bound DOPG-modulated receptor conformations, obtaining similar fits to that observed in the G$_s$ protein-bound crystal structure [25]. In all aspects, these MD-generated active-like receptor states are remarkably consistent with the crystallized NECA-bound active state [24, 25]. Furthermore, these results are reproducible across different MD simulations with different agonists, revealing consistent cooperativity between agonist and DOPG lipids in the A2aR activation process. It also supports the notion that anionic phospholipids are crucial for activation of class A GPCRs in general [49, 50, 53]. Intriguingly, it is possible to differentiate between two distinct active-like receptor conformations, which are dependent on slightly different TM6 and TM7 orientations. These observations are supported by recent NMR studies where A2aR appeared able to have more than one active state, although their precise conformational details were not revealed [61]. Most notably across our MD simulations, protein-DOPG interactions assist receptor activation through enhanced stabilization of bound agonist in the orthosteric pocket. This effect is strongest for the endogenous agonist

adenosine, which is able to form more stable protein-ligand H-bonds as a result but it is also apparent with NECA, thus providing an explanation of how ligand efficacy might be dependent on lipid environment [65, 77]. Achieving a stable binding pose of agonist appears a precondition for obtaining the fully active receptor state and is a likely consequence of more stable protein-ligand interactions with TM6, TM7, as well as ECL2. These observations are similar to experimental data reported in β2AR, where DOPG lipids improve the binding of agonists and activation of the receptor [49]. In this context, it is logical that DOPG lipids stabilize the binding of agonists in A2aR.

The cooperative effects between ligands and lipids are the underlying principle of the present study. The cooperativity described in this work corresponds to allosteric interactions in the broad sense, i.e. specific protein-lipid interactions affecting the binding stability of the agonist and binding of the agonist focusing the functional effect of protein-lipid interactions, inducing specific conformational changes in the receptor. Most notably, DOPG lipids accelerate the activation process of an agonist-bound receptor, which in a microsecond timeframe enables full receptor activation to occur more consistently but does not always guarantee it. Interestingly, when comparing A2aR activation with that of CB1 [53], it is striking that activation of the former only proceeds with PG lipids while activation of the latter proceeds in both PG and PC lipid environments, albeit faster and more reliably with PG. This may be because these two receptors show different levels of activity. In particular, CB1 is known to have high constitutive activity and may be easier to activate relative to other GPCRs [8]. For example, as previously noted in the core of CB1, the semi-conserved residue L$^{6.44}$ may confer higher TM6 flexibility, which may allow for easier receptor conformational change [53]. The equivalent residue in A2aR is F242$^{6.44}$, which as a bulkier amino acid may enhance hydrophobic interactions in the receptor core that help to stabilize the inactive state. This could be one reason why activation of A2aR is observed more scarcely in the microsecond timescale relative to CB1 and why protein-lipid interactions appear to be so important for A2aR [77]. Presumably, if our MD simulations were extended into the millisecond time range, then we might observe full A2aR activation in PC membranes. However, this is currently an untestable hypothesis. Irrespective of this, we consider our findings to be sufficiently compelling and potentially enable a better understanding of receptor activation, agonist efficacy, and protein-lipid positive allosteric modulation in A2aR, especially as its observed dynamics broadly mirror those of other class A GPCRs, such as muscarinic M2, CB1 and opsin [43, 53, 117].

## Methods

### A2aR modelling

The high-resolution crystal structure of the antagonist-bound inactive state of human adenosine A2a receptor (A2aR) (PDB entry: 4EIY) [29] was selected and co-crystallized ligands, waters and ions were removed. By using CHIMERA software [118], crystallographic missing atoms were added (on residues Q148, E161, R220, R293) and non-native residues were removed or converted to native (at N-terminus: G-1 and A0 deleted, P1 substituted for M) in order to obtain a *wt* receptor sequence. In addition, the non-native fusion protein located between L208 (on TM5) and E219 (on TM6) was excised and the crystallographic missing intracellular loop 3 (ICL3) was modelled (residues 209 to 218) by basing it upon the equivalent region of thermostabilized A2aR crystal structure (PDB entry: 3PWH) [79] using MODELLER v9.14 [119].

In order to validate MD generated conformations, the intermediate adenosine-bound (PDB entry: 2YDO), intermediate NECA-bound (PDB entry: 2YDV) [93] and NECA-bound fully active (PDB entry: 6GDG) [25] A2aR crystal structures were utilized. Non-native residues

were converted to wt in both structures and crystallographic missing ICL3 and extracellular loop 2 (ECL2) were completed in each respective structure using relevant A2aR crystal structures: PDB entries 3PWH [79] or 5G53 [24] as templates with MODELLER software [119].

The structures of adenosine and NECA were retrieved from their respective crystal structures of thermostabilized adenosine/NECA-bound A2aR (PDB entry: 2YDO/2YDV) [93]. As these thermostabilized receptor states are in a similar conformation to our utilized inactive state (PDB entry: 4EIY) [29], docking of adenosine and NECA was performed by firstly superimposing receptor structures (PDB entries: 2YDO or 2YDV onto 4EIY) with CHIMERA [118] and then, secondly, by transferring the coordinates of the agonist from one to the other whilst avoiding steric conflicts where possible. Finally, in order to optimize protein-ligand contacts, the resulting complexes were energy-minimized in the AMBER-14SB force-field [120] with CHIMERA [118].

## Molecular dynamics (MD) simulations

Eight all-atom MD systems were constructed using CHARMM-GUI [121], consisting of A2aR in an inactive state (as described above) with or without bound adenosine or NECA embedded in DOPG, DOPC or POPC homogeneous lipid bilayers (80 Å x 80 Å), respectively. The position of the receptor in the membrane(s) is as reported by the OPM database for A2aR PDB entry: 4EIY [29]. Each receptor-membrane system was solvated with TIP3P water molecules above and below the membrane, and an overall concentration of 0.3 M K$^+$Cl$^-$ with ratio of positive/negative ions adjusted accordingly (automatically during CHARMM-GUI system setup) in order to maintain net zero charge in each system. On the protein, eight residues were protonated according to previously published MD simulation protocols specific for A2aR [44, 72, 74, 78, 122], as well as being consistent with protocols used for other homologous class A GPCRs [47, 53, 95, 117, 123]: E13$^{1.39}$, D52$^{2.50}$ (both receptor core), D101$^{3.49}$, E212$^{5.73}$, E219$^{6.21}$, E228$^{6.30}$, H230$^{6.32}$, E294$^{8.49}$ (all intracellular) (with Ballesteros numbering [99] as superscript, which indicates relative residue position along each TM helix). Membrane, water and protein parameters were generated according to the CHARMM36 force-field [124] and adenosine parameters were generated according to CGenFF v1.0.0 [125]. MD simulations were performed with ACEMD software [126] on specialized GPU-computer hardware. Briefly, each receptor-membrane system was equilibrated for 28 ns at 300 K (Langevin thermostat) and 1 atmosphere (Berendsen barostat) using a 4 fs time-step and electrostatics cut-off of 9.0 Å. During the initial 8 ns of equilibration, protein (and ligand) heavy atoms were harmonically restrained and progressively released over 2 ns steps. During the final 20 ns of equilibration, no restraints were applied. For each receptor-membrane system, a production run of 2 μs was performed without restraints under the same conditions, with second, third and fourth replicate simulations executed in each case to verify observations (in case of POPC systems, only a second replicate was executed). This constitutes a total production simulation time of 56 μs across the eight systems.

## MD simulation analysis

Structural comparison was carried out for receptor conformations sampled from MD simulations against A2aR crystal structures: fully active (PDB entry: 6GDG) [25], intermediate (PDB entries: 2YDO and 2YDV) [93], or inactive state (PDB entry: 4EIY) [29]. In each MD simulation trajectory, in order to assess membrane effects on receptor conformation, root mean square deviation (RMSD) of Cα atoms of TM domain (TM helices 1–7), as well as TM6 and ECL2 by themselves, were calculated with VMD software [127] v1.9.2. To further validate receptor state, multiple internal measurements were made including: χ1 dihedral angle of

W246$^{6.48}$, vertical movement of TM3 and ECL2 according to distance offset in the Z axis (perpendicular to the membrane) from centre of mass of each to centre of mass of TM domain, distance between Cα atoms of residue pairs: R102$^{3.50}$ and E228$^{6.30}$, R102$^{3.50}$ and Q207$^{5.68}$, and R102$^{3.50}$ and Y288$^{7.53}$, representing proximity between helices: TM3-TM6, TM3-TM5 and TM3-TM7, respectively. The following lipid measurements were calculated: i) protein-lipid distance between residues R102$^{3.50}$ (sidechain terminal nitrogen atoms) or H230$^{6.32}$ (sidechain centre-of-mass) and closest lipid phosphate group (centre-of-mass) using PLUMED v2.4 [128]; (ii) membrane thickness as average distance from lower to upper-leaflet phosphorus atoms as per default settings of MEMBPLUGIN [129] within VMD [127]. For the ligand: (i) protein-ligand π–π stacking was calculated as function of distance between centre of mass of aromatic ring of F$^{45.52}$ and adenine ring of adenosine or NECA (applied criteria for interaction: 3.5 Å); ii) "Hydrogen Bonds" function within VMD was used to analyse protein-ligand H-bond occupancies (applied criteria of donor-acceptor distance: 3.5 Å and 60˚ angle); (iii) ligand conformational change in terms of RMSD with respect to initial position; and (iv) ligand positional stability expressed in terms of root mean square fluctuation (RMSF) were made within VMD. Internal water molecule density was measured by the function "Volmap" within VMD using default options. Criteria for water-mediated interactions in the receptor were defined by two residues forming hydrogen bonds with the same water molecule or a pair of hydrogen bonded water molecules. All analytical plots were generated using Matplotlib version 3.0.0 [130].

## Protein-Protein docking

As validation of receptor state, co-crystallized G$_s$α protein (PDB entry: 6GDG) [25] was docked to the intracellular side of selected MD-generated A2aR conformations (as well as re-docked into the original fully active A2aR crystal structure as a control). The Rosetta online server (ROSIE) was used for protein-protein docking [107] with the following protocol: (i) receptor conformation taken from simulation of its respective MD simulation and superimposed over the active crystal structure of A2aR containing its G$_s$α protein (PDB entry: 6GDG) [25], (ii) the original crystallized receptor removed from the complex, (iii) the structure of G$_s$α moved 3.0 Å away from the MD-generated receptor conformation so that there are no steric clashes and clear space is apparent between both proteins, (iv) protein-protein docking is initiated. As ICL3 is long and potentially highly flexible, it was removed prior to protein-protein docking (during step iii) so as not to create unavoidable steric conflicts with G$_s$α during docking (ROSIE is not able to move backbone of loops). However, all other loops and receptor structural elements were maintained.

## Supporting information

**S1 File. Details of the simulations.** This zip file contains a README document with data and software description as well as topology, scripts and input files within the corresponding generated folders.
(ZIP)

**S1 Fig. Docking of NECA in the inactive crystal structure of adenosine A2a receptor (A2aR).** A) Molecular structure of NECA. Comparison of B) co-crystallized NECA (lime) in agonist-bound A2aR crystal structure (PDB entry: 2YDV, light green), and C) docked NECA (magenta) in the inactive crystal structure of A2aR (PDB entry: 4EIY, pink). Selected residues participating in ligand binding are displayed. Extracellular loop (ECL) 2 and transmembrane

(TM) helices 5–7 are labelled.
(TIF)

**S2 Fig. Structural comparison between A2aR intermediate and inactive crystal structures.**
A) structural superposition of the intermediate adenosine-bound crystal structure (PDB entry:
2YDO, light green) on the inactive-state crystal structure (PDB entry: 4EIY, pink). B) Comparative positioning of residue L$^{3.43}$ located on TM3 and rotameric state of W$^{6.48}$ on TM6. C)
Intracellular distance between residues R$^{3.50}$ and Y$^{7.53}$ on TM3 and TM7 (indicated by dashed lines) before/after receptor conformational change. D) Distance between residues R$^{3.50}$ and Q$^{5.68}$ (indicated by dashed lines). E) Partial separation of ionic-lock residues R$^{3.50}$ and E$^{6.30}$ on TM3 and TM6 (indicated by dashed lines). Relevant structural features are labelled: extracellular loops (ECL) 1, 2 and 3, and transmembrane (TM) helices 1–3, 5–7.
(TIF)

**S3 Fig. Structural comparison between A2aR active and inactive crystal structures.** A)
structural superposition of the active-state crystal structure (PDB entry: 6GDG, brown) on the inactive-state crystal structure (PDB entry: 4EIY, pink). B) Proposed scheme of activation for A2aR, including rotation and upwards axial movement of TM3, outwards movement of TM5, rotation plus outward movement of TM6, and inwards movement of TM7. C) Comparative positioning of residue L$^{3.43}$ located on TM3 and rotameric state of W$^{\mathbf{6.48}}$ on TM6. D) Intracellular conformational change of TM5 with increased separation (indicated by dashed lines) between residues R$^{3.50}$ and Q$^{5.68}$ after receptor activation. E) Intracellular comparison of distance between residues R$^{3.50}$ and Y$^{\mathbf{7.53}}$ after receptor activation (indicated by dashed lines). F)
Intracellular conformational change of TM6 and separation (indicated by dashed lines) of ionic-lock residues R$^{3.50}$ and E$^{6.30}$ after receptor activation. Relevant structural features are labelled: intracellular loop (ICL) 2, extracellular loops (ECL) 1, 2 and 3, and transmembrane (TM) helices 1–3, 5–7.
(TIF)

**S4 Fig. Comparison of conformational change of extracellular loop 2 (ECL2) in MD simulations of A2aR.** A) RMSD of ECL2 from the starting inactive A2aR crystal structure (PDB entry: 4EIY). B) Vertical movement of ECL2 along Z-axis (containing residues: G142-A173).
MD simulations are performed in quadruplicate, with or without bound adenosine (ADN) in DOPC or DOPG homogeneous membranes.
(TIF)

**S5 Fig. Conformational change of helix bundle of A2aR in MD simulations.** A) RMSD of helices 1–7 from the inactive crystal structure (PDB entry: 4EIY) and B) with respect to the active crystal structure of A2aR (PDB entry: 6GDG). MD simulations are performed in quadruplicate, with or without bound adenosine (ADN) in DOPC or DOPG homogeneous membranes.
(TIF)

**S6 Fig. TM6 conformational change of A2aR in MD simulations.** A) RMSD from the starting inactive A2aR crystal structure (PDB entry: 4EIY) and B) with respect to the active A2aR crystal structure (PDB entry: 6GDG). MD simulations are performed in quadruplicate, with or without bound adenosine (ADN) and in DOPC or DOPG homogeneous membranes.
(TIF)

**S7 Fig. Assessment of conformational change along TM3 during MD simulations of A2aR.**
A) RMSD of residue L3.43 on TM3 compared to the inactive crystal structure (PDB entry:
4EIY) and B) assessment of vertical movement of TM3 along Z-axis. MD simulations are

performed in quadruplicate with or without bound adenosine (ADN) and in DOPC or DOPG homogeneous membranes.
(TIF)

**S8 Fig. Assessment of rotameric conformational change of residue W6.48 on TM6.** A) W246[6.48] rotameric switch starting from *gauche*(-) (285˚) (belonging to replica #2 from 1.7 μs in APO embedded in DOPC, in magenta) to trans (180˚) during MD simulation replica #4 from 1.8 μs in DOPG with bound adenosine (in green). B) χ1 dihedral angle of residue W246[6.48] over time. MD simulations are performed in quadruplicate, with or without bound adenosine (ADN) in DOPC or DOPG homogeneous membranes.
(TIF)

**S9 Fig. Comparison of TM3-TM5 inter-helical distance in MD simulations of A2aR.** Distance between TM3-TM5 is measured between Cα atoms of R102[3.50] and Q207[5.68]. MD simulations are performed in quadruplicate, with or without bound adenosine (ADN) in DOPC or DOPG homogeneous membranes.
(TIF)

**S10 Fig. Alternative binding pose of adenosine in A2aR in DOPC membrane.** A) Superposition of the intermediate crystal structure of A2aR (PDB entry: 2YDO, light green) and an MD-generated conformation achieved within a DOPC membrane bound to adenosine (in blue, belonging to replica #1 at 1.9 μs) showing B) and C) ligand atoms as spheres and selected residues making protein-ligand interactions as sticks. Intracellular loop (ICL) 3, extracellular loop (ECL) 2, and transmembrane (TM) helices 1–3 and 5–7 are labelled.
(TIF)

**S11 Fig. Assessment of key protein-ligand interactions in MD simulations of adenosine-bound A2aR embedded in DOPC membrane.** A) Left: distance of F[45.52] with respect to ribose moiety of adenosine (ADN). Right: frequency (%) of protein-ligand π-π stacking (within range of 0.0 to 4.0 Å) over 2 μs. B) Evaluation of protein-ligand H-bond distances formed by residues: N253[6.55], E169[45.53], H278[7.43], S277[7.42] (N—O or O—O). C) Mean protein-ligand interactions (%) and protein-ligand H-bond occupancies per replica (%) for selected residues.
(TIF)

**S12 Fig. MD simulation data of adenosine A2a receptor (A2aR) with or without bound adenosine (ADN) starting from the inactive receptor crystal structure (PDB id: 4EIY) in a POPC membrane.** Top row: RMSD and conformational fluctuation (RMSF) of bound adenosine ligand; second row: TM3-TM7 and ionic lock (TM3-TM6) inter-helical distances; third row: RMSD of whole TMD (TMs 1–7) or only TM6; fourth row: RMSD compared to active crystal structure (PDB id: 6GDG) of whole TMD (TMs 1–7) or only TM6; fifth row: χ1 dihedral angle of W246[6.48] on TM6 starting from *gauche*(-) crystal position (285˚), and vertical movement of extracellular loop 2 (ECL2); bottom row: vertical movement of TM3 and RMSD of ECL2. MD simulations are performed in duplicate in POPC homogeneous membranes.
(TIF)

**S13 Fig. Protein-lipid interaction between closest DOPG molecule and residue H230 6.32 in MD simulations of A2aR with or without bound adenosine.** A) Residues H230[6.32] and K233[6.35] on TM6 of A2aR in apo state (orange) and B) A2aR with bound adenosine (green) interacting with DOPG lipid (gold). Histidine-lipid interaction distances over time in four replicas of A2aR in DOPG membrane in C) apo state and D) adenosine-bound (ADN), respectively.
(TIF)

**S14 Fig. Intermediate conformation of the apo state of A2aR in a DOPG membrane.** A) Comparison of the intermediate crystal structure of A2aR (PDB entry: 2YDO, light green) and an MD-generated apo conformation achieved within a DOPG membrane (in red, belonging to replica #2 at 1.6 μs) showing B) and C) selected residues delineating the orthosteric pocket. Intracellular loop (ICL) 3, extracellular loop (ECL) 2, and transmembrane (TM) helices 1–3, 5–7 are labelled.
(TIF)

**S15 Fig. Assessment of key protein-ligand interactions in MD simulations of adenosine-bound A2aR embedded in DOPG membrane.** A) Left: distance of F$^{45.52}$ with respect to ribose moiety of adenosine (ADN). Right: frequency (%) of protein-ligand π-π stacking (within range of 0.0 to 4.0 Å) over 2 μs. B) Evaluation of protein-ligand H-bond distances formed by residues: N253$^{6.55}$, E169$^{45.53}$, H278$^{7.43}$, S277$^{7.42}$ (N—O or O—O). C) Mean protein-ligand interactions (%) and protein-ligand H-bond occupancies per replica (%) for selected residues.
(TIF)

**S16 Fig. State-dependent water-mediated polar network across the ZM241385-bound and adenosine-bound A2aR crystal structures.** The water network retrieved from: A) the inactive crystal structure (PDB entry: 4EIY, pink) and B) the intermediate crystal structure (PDB entry: 2YDO, light green). Residues and ligands are shown as sticks and water molecules are shown as red spheres.
(TIF)

**S17 Fig. Alternative active-like conformational state of A2aR generated in an MD simulation with bound adenosine in a DOPG membrane.** A) Comparison of an MD-generated conformation of A2aR bound to adenosine within a DOPG membrane (in green, belonging to replica #4 at 1.8 μs) with the active crystal structure of A2aR (brown, PDB entry: 6GDG) showing B) and C) ligand atoms as spheres and residues making protein-ligand interactions as sticks. Intracellular loop (ICL) 3, extracellular loop (ECL) 2, and transmembrane (TM) helices 1–3, 5–7 are labelled.
(TIF)

**S18 Fig. Comparison of two active-like receptor conformations generated in MD simulations of A2aR with bound adenosine in a DOPG membrane.** A) Comparison of an MD-generated conformation of A2aR bound to adenosine within DOPG belonging to replica #2 (at 1.6 μs, green) with respect to replica #4 (at 1.8 μs, olive green) showing B) and C) ligand atoms as spheres and residues making protein-ligand interactions as sticks. Intracellular loop (ICL) 3, extracellular loop (ECL) 2, and transmembrane (TM) helices 1–3, 5–7 are labelled.
(TIF)

**S19 Fig. MD simulation data of adenosine A2a receptor (A2aR) with bound NECA starting from the inactive receptor crystal structure (PDB id: 4EIY) in a DOPC membrane.** Top row: RMSD and conformational fluctuation (RMSF) of bound NECA ligand; second row: TM3-TM7 and ionic lock (TM3-TM6) inter-helical distances; third row: RMSD of whole TMD (TMs 1–7) or only TM6; fourth row: RMSD compared to active crystal structure (PDB id: 6GDG) of whole TMD (TMs 1–7) or only TM6; fifth row: χ1 dihedral angle of W246$^{6.48}$ on TM6 starting from *gauche*(-) crystal position (285˚), and vertical movement of extracellular loop 2 (ECL2); bottom row: vertical movement of TM3 and RMSD of ECL2. MD simulations are performed in quadruplicate in DOPC homogeneous membranes.
(TIF)

**S20 Fig. MD simulation data of adenosine A2a receptor (A2aR) with bound NECA starting from the inactive receptor crystal structure (PDB id: 4EIY) in a DOPG membrane.** Top row: RMSD and conformational fluctuation (RMSF) of bound NECA ligand; second row: TM3-TM7 and ionic lock (TM3-TM6) inter-helical distances; third row: RMSD of whole TMD (TMs 1–7) or only TM6; fourth row: RMSD compared to active crystal structure (PDB id: 6GDG) of whole TMD (TMs 1–7) or only TM6; fifth row: $\chi 1$ dihedral angle of $W246^{6.48}$ on TM6 starting from *gauche*(-) crystal position (285˚), and vertical movement of extracellular loop 2 (ECL2); bottom row: vertical movement of TM3 and RMSD of ECL2. MD simulations are performed in quadruplicate in DOPG homogeneous membranes.
(TIF)

**S21 Fig. Assessment of key protein-ligand interactions in MD simulations of NECA-bound A2aR embedded in DOPC membrane.** A) Left: distance of $F^{45.52}$ with respect to ribose moiety of NECA. Right: frequency (%) of protein-ligand π-π stacking (within range of 0.0 to 4.0 Å) over 2 μs. B) Evaluation of protein-ligand H-bond distances formed by residues: $N253^{6.55}$, $E169^{45.53}$, $H278^{7.43}$, $S277^{7.42}$ (N—O or O—O). C) Mean protein-ligand interactions (%) and protein-ligand H-bond occupancies per replica (%) for selected residues.
(TIF)

**S22 Fig. Assessment of key protein-ligand interactions in MD simulations of NECA-bound A2aR embedded in DOPG membrane.** A) Left: Distance of $F^{45.52}$ with respect to ribose moiety of NECA. Right: frequency (%) of protein-ligand π-π stacking (within range of 0.0 to 4.0 Å) interaction over 2 μs. B) Evaluation of protein-ligand H-bond distances formed by residues: $N253^{6.55}$, $E169^{45.53}$, $H278^{7.43}$, $S277^{7.42}$ (N—O or O—O). C) Mean protein-ligand interactions (%) and protein-ligand H-bond occupancies per replica (%) for selected residues.
(TIF)

**S23 Fig. Protein-lipid allosteric interaction with the ionic-lock in MD simulations of NECA-bound A2aR in DOPG membrane.** A) Electrostatic interaction between ionic-lock residue $R102^{3.50}$ of A2aR (green) from an intracellular viewpoint and a DOPG lipid, which intrudes between TM6 and TM7 (snapshot belonging to replica #1 at 1.7 μs). B) Protein-lipid interaction distance over time between $R102^{3.50}$ sidechain and lipid phosphate group in four replicas of NECA-bound A2aR in DOPG membrane.
(TIF)

**S24 Fig. Frequency distribution of receptor conformations formed during different MD simulations with and without bound adenosine in two different membranes (DOPC or DOPG) according to two different inter-helical distances.** A) Population of receptor conformations according to distance between residues $R^{3.50}$ and $Y^{7.53}$ (TM3-TM7), and (B) between ionic lock residues $R^{3.50}$ and $E^{6.30}$ (TM3-TM6). Vertical black lines indicate values of inactive (PDB entry: 4EIY), intermediate (PDB entry: 2YDO) and active (PDB entry: 6GDG) crystal structures.
(TIF)

**S25 Fig. Frequency distribution of receptor conformations (according to two inter-helical distances) formed during MD simulations with bound NECA in two different membranes (DOPC or DOPG) or with/without bound adenosine in a POPC membrane.** First and second rows: population of receptor conformations according to distance between $^{R3.50}$ and $^{Y7.53}$ (TM3-TM7); third and fourth rows: according to distance between ionic lock residues $^{R3.50}$ and $E^{6.30}$ (TM3-TM6). Vertical black lines indicate values of inactive (PDB entry: 4EIY),

intermediate (PDB entry: 2YDO) and active (PDB entry: 6GDG) crystal structures.
(TIF)

**S26 Fig. Boxplots of receptor characteristics from MD simulations of adenosine A2a receptor (A2aR) in apo or with bound adenosine (ADN) or NECA in three different membranes.** Top row: inter-helical distances between: residues R$^{3.50}$ and Y$^{7.53}$ (TM3-TM7), and ionic lock residues R$^{3.50}$ and E$^{6.30}$ (TM3-TM6); bottom row: vertical movements of extracellular loop 2 (ECL2) and TM3, respectively. MD simulations in DOPG or DOPC were performed in quadruplicate. MD simulations in POPC were performed in duplicate.
(TIF)

**S1 Table. Adenosine A2a receptor crystal structures.**
(TIF)

**S2 Table. Comparison of A2aR crystal structure distances.** Comparison of TM3-TM6, TM3-TM5 and TM3-TM7 inter-helical distances in active, intermediate and inactive crystal states.
(TIF)

**S3 Table. Evaluation of Gαs protein docking.** Comparison of best docking quality of Gα$_s$ protein into inactive, intermediate and active crystals structures, and different MD-generated conformations of A2aR achieved under different conditions and performed in quadruplicate.
(TIF)

## Author Contributions

**Conceptualization:** James A. R. Dalton, Jesús Giraldo.

**Formal analysis:** Agustín Bruzzese, James A. R. Dalton, Jesús Giraldo.

**Funding acquisition:** Jesús Giraldo.

**Investigation:** Agustín Bruzzese, James A. R. Dalton, Jesús Giraldo.

**Methodology:** Agustín Bruzzese, James A. R. Dalton, Jesús Giraldo.

**Resources:** Jesús Giraldo.

**Supervision:** James A. R. Dalton, Jesús Giraldo.

**Writing – original draft:** Agustín Bruzzese, James A. R. Dalton.

**Writing – review & editing:** James A. R. Dalton, Jesús Giraldo.

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
