## [Decision Letter · Decision Letter 0]

24 Jan 2020

Dear Dr. Giraldo,

Thank you very much for submitting your manuscript "Insights into Adenosine A2A receptor activation through cooperative modulation of agonist and allosteric lipid interactions" for consideration at PLOS Computational Biology.

As with all papers reviewed by the journal, your manuscript was reviewed by members of the editorial board and by several independent reviewers. In light of the reviews (below this email), we would like to invite the resubmission of a significantly-revised version that takes into account all the reviewers' comments.

We cannot make any decision about publication until we have seen the revised manuscript and your response to the reviewers' comments. Your revised manuscript is also likely to be sent to reviewers for further evaluation.

Sincerely,

Bert L. de Groot

Associate Editor

PLOS Computational Biology

Arne Elofsson

Deputy Editor

PLOS Computational Biology

Reviewer's Responses to Questions

**Comments to the Authors:**

Reviewer #1: The manuscript "Insights into adenosine A2A receptor activation through cooperative modulation of agonist and allosteric lipid interactions" by Bruzzese et al. describes the lipid-dependent activation of the receptor. The authors have performed 2 microsecond atomistic MD simulations on 4 different systems: the apo and ligand bound inactive structure of the receptor in DOPC and DOPG bilayers, with 4 replicates for each system in consideration. Based on these simulations, the authors suggest that lipid effects and ligand binding affect receptor activation states. There have been several papers using simulations analyzing the A2A activation and the authors need to highlight salient features of their own work better. I have several comments that need to be taken into consideration to strengthen the manuscript.

1. Is the entry of a DOPG molecule into the lumen of the receptor an artifact of the system setup and the micro-system with large electrostatic effects (charged lipids etc.)? Bilayers of only PG lipids are quite difficult to handle experimentally due to the large electrostatic repulsion between the lipids. The authors need to validate that this is not an artifact of their system set up and simulations. For instance what are the counter ions used? I would recommend using a PC/PG mixed bilayer to substantiate their findings.

2. One major weak point that I see is that the lipid is suggested to act as an allosteric modulator, without showing specific (converged) binding sites and discussing allosteric communication pathways. Otherwise these could be simple non-specific "environment effects". Calling it an allosteric effect implies a specific allosteric binding and long-range communication pathways from that site to either ligand binding or G-protein coupling site. In this regard, the allosteric behavior would need to be more rigorously shown.

3. Is DOPC is a correct membrane control ? The high fluidity and curvature of the system is perhaps not a good control of a"generic membrane". In fact the authors at one point write that it can be compared to POPC, a true "representative" lipid. This statement should be modified and the authors to justify the use of DOPC/DOPG.

4. I would suggest that in addition to presenting the results as time evolution (of the distances), they are represented as population distributions to have a better understanding the contribution of different conformational states.

5. The authors have extensively used RMSDs as their major indicator to identify conformational states at the start of the paper. I suggest authors use other collective variables (such as what they have calculated later, ionic lock, TM6-TM2 distance etc.) to this more robust.

6. Statistical and sampling errors needs to be presented for all plots.

7. There are a large number of papers that have identified cholesterol binding sites on this receptor. Since cholesterol is an important constituent of the membrane where the receptor resides, the authors should discuss their results in the context of cholesterol binding. If specific cholesterol binding sites are that critical for the receptor, wouldn't these results be misleading? Perhaps a mixed bilayer (with cholesterol) be more relevant. In the absence of such simulations, the authors need to discuss these points better. There have been reports of cholesterol in the receptor lumen for this receptor. Does it replace DOPG in the lumen or is it cooperative binding?

8. The authors need to provide more details on the MD parameters used: pressure coupling, temperature coupling, handling of electrostatics etc. for completeness. If there are word restrictions, perhaps the details can be provided in the Supplementary.

9. A homogeneous negatively charged bilayer would introduce electrostatic artifacts in the system and could largely affect the protein behavior. How is this taken care of - since the system does not appear to have neutralizing counter-ions added. Similarly, where are these lipids actually located in the bilayer- which leaflet of the bilayer. In this regard, are the results consistent with the known lipid abundance?

10. Line 220 : “Moreover, the distance between TM3 and TM7 is decreased, most noticeably between intracellular residues R1023.50 and Y2887.53, while the distance between TM3 and TM5 is increased, exemplified by residues R1023.50 and Q2075.68 (S1C-E Fig) (S2D-F Fig) (S1 Table).”

>> The sentence could be better discussed with the actual values of these distance to understand the magnitude of change in the distances.

>> typo in the table reference. It should be SI Table 2.

>> please check the consistency of the BW numbering as a superscript throughout the manuscript and for figure legends too.

11. Line 223 : “The main difference between intermediate and active crystal structures can be observed in the distance between TM3 and TM6 and intracellular ionic-lock residues R1023.50 and E2286.30, respectively.”

>> all these observations seem qualitative, referring to the actual value of the distance while describing in the results would be helpful.

12. Line 225 : “Both crystals have TM6 oriented outwards compared with the inactive crystal structure. However, TM3- TM6 distance reaches 18.5 Å in the active, but only 9.9 Å in the intermediate crystal 228 structure (S1E and S2F Fig) (S2 Table)”

>> It’s 18.3Å, as mentioned in the table by authors - I guess 0.2Å makes a difference? Or put it as approx.

>> Please refer to the table in the SI consistently: Table SI 2

13. Line 244 : “Over the course of these replicas, ECL2 continues to fluctuate up to a maximum of 8.6 Å and 10.9 Å RMSD, respectively (S3A Fig).”

>> It appears that except for DOPC-apo simulations (replicate #2 and #3 ), all others - even for DOPG are comparable, indicating there is no difference in simulations with different bilayers.

Why do you think that is? Is it possible it is not well equilibrated? or converged ? Statistical significance should be provided.

14. Line 247 : “However, in other replicas, ECL2 shows less conformational change (average RMSD across replicas #1 and #4 of 2.4 Å ±0.7 S.D.)” (Figure S3A)

>> In figure S3B, there appears to be no line corresponding to the active crystal structure. Could the authors clarify?

15. Line 248 : Despite these conformational fluctuations of ECL2, little conformational change is observed in the receptor as a whole across all four replicas with an average RMSD of 2.5 Å ±0.6 S.D. (S4A Fig) or in key helices such as TM3 (average Z-axis displacement of 0.2 Å ±0.6 S.D., Fig 3G)

>> >> Here I feel, inclusion of Helix 8 is not necessary since TM helices are the main hub of structural changes. Also, helix 8 is a membrane associated helix and not TM helix.

>> In case of Figure S4A and S4B, RMSDs in A) panel are higher than that of the panel B). Does this indicate the structures are more similar to active crystal structure than the inactive in both the bilayers? This needs to be discussed.

16. Line 265 : “Despite some of these observed differences, TM7 regularly revisits its inactive crystal conformation across all four replicas, suggesting this state is at least meta-stable and in line with the receptor remaining inactive (Fig 3A, D)”

>> A population analysis of this distance variation might represent the results better and actually show the increased presence of this conformation in the total ensemble sampled.

>> Also refer to figure 11

17. Line 286 : “In replicas #3 and #4, adenosine achieves a stable binding mode in the orthosteric pocket between helices TM3, TM6 and TM7, which is consistent with its crystallized pose in the A2aR intermediate crystal structure [93].”

>> Please refer to S10 here

18. Line 289 : “At the same time, N2536.55 forms a stable protein-ligand H-bond with average occupancy of 95.0% (±5.4 S.D.)”

>> Please clarify the residue number N6.55 or 6.33 and check S13 legend and figure.

19. Please label the helices in Figure 6. Are the thickness plots averages over one set or 4 replicates ?

20.Line 323 : “Regarding key protein-ligand interactions, N2536.33 located on TM6 and S2777.42 324 and H2787.43 located on TM7 make H-bonds with adenosine with average occupancies of 92.1% (±7.0 S.D.), 49.5% (±25.2 S.D.) and 62.9% (±16.1 S.D.) (S9B Fig)”

>> It should be Figure S9C not S9B

>> also please provide error bars on the plot

21. Line 332 : “This precedes the gradual separation of the intracellular end of TM6 away from TM3 by a maximum of 5.1 Å, which starts at a distance of 8.3 Å and increases up to 13.4 Å within 0.7 μs of the same three replicas (Fig 4 G).”

>> How do we know this isn’t a one off event ? Could the authors comment on the statistical significance?

22. “Our MD simulations suggest that ECL2 is likely to be more stabilized in a higher position in a DOPG environment rather than in DOPC, which appears to assist conformational change in TM3 as well as enhancing the stability of bound adenosine. On a speculative level, this influence of lipids on ECL2 may provide a structural explanation for the observation that GPCRs exhibit different ligand efficacies in different cell lines [108, 110]”

>> Why is ECL2 stabilized in a higher position? If there a DOPG lipid binding at that site? Or are there long-range effects. This appears to be a central part of the paper and needs to be discussed better.

23. Regarding the lipid interactions - I feel that a better characterization is required. Is the lipid always there? Are there other lipid binding sites?

24. Line 491 : “In addition, adenosine-bound A2aR in DOPG displays the same four protein-lipid allosteric interaction hot-spots as seen with the apo state in DOPG: at TM1- H8, TM3-4, TM5 and TM6 (Fig 6, S11 Fig).”

>> In Figure S11, do the the snapshots belong to apo-state or ligand bound state? Could the authors update the figure legend?

25. Line 776 : “Interestingly, when comparing A2aR activation with that of CB1 [53], it is striking that activation of the latter proceeds more consistently and in different membrane types. This This may be a feature of the type of agonist used in each study, for example a smaller and potentially weaker endogenous agonist here in A2aR, and a larger potentially stronger synthetic agonist in CB1”.

>> If this statement is considered, then the lipid induced effect observed here may not be a universal phenomenon for this receptor. In that case, the title of the paper should explicitly mention the use of adenosine (natural agonist) as an agonist for the study.

Reviewer #2: In this manuscript, Bruzzese at all., performed MD simulations of the A2A receptor inactive conformation in adenosine-bound and apo forms in DOPG and DOPC and observed transition to intermediate active receptor conformations in DOPC bound to adenosine and DOPG in apo and adenosine-bound forms. In the case of the adenosine-bound receptor form in DOPG they got a conformation that binds the Gs protein in docking studies. They showed that DOPG has positive allosteric effects into active conformation transition. The study is built on the available crystal structures of A2A receptors in inactive, active and intermediate conformations and uses a number of characteristic structural features of the receptor active state to monitor conformational changes. The manuscript is clearly written and the claims are convincingly supported. The work is recommended for publication after addressing the following comments:

1. The quality of the images in the main text is really poor, please check.

2. Page 17, line 353 ‘are these’ should be corrected. This manuscript would benefit from review by a professional editing service.

3. Figure 6. Good to label helices to link with the discussed text on page 19-20.

4. Page 28-29, units should be provided for interface docking score.

5. Figure S1, S2. Dashed lines are shown to indicate the distances but the values are not shown. It is hard to appreciate the differences in distances.

6. Figure 11S should be in the text.

**Have all data underlying the figures and results presented in the manuscript been provided?**

Reviewer #1: Yes

Reviewer #2: Yes

PLOS authors have the option to publish the peer review history of their article (what does this mean?). If published, this will include your full peer review and any attached files.

Reviewer #1: Yes: D Sengupta

Reviewer #2: No
---

## [Decision Letter · Decision Letter 1]

23 Mar 2020

Dear Dr. Giraldo,

We are pleased to inform you that your manuscript 'Insights into adenosine A2A receptor activation through cooperative modulation of agonist and allosteric lipid interactions' has been provisionally accepted for publication in PLOS Computational Biology.

Best regards,

Bert L. de Groot

Associate Editor

PLOS Computational Biology

Arne Elofsson

Deputy Editor

PLOS Computational Biology

Reviewer's Responses to Questions

**Comments to the Authors:**

Reviewer #1: The manuscript has substantially improved and would be a good addition to the field.

**Have all data underlying the figures and results presented in the manuscript been provided?**

Reviewer #1: Yes

PLOS authors have the option to publish the peer review history of their article (what does this mean?). If published, this will include your full peer review and any attached files.

Reviewer #1: Yes: Durba Sengupta

---

## [Editor Report · Acceptance letter]

3 Apr 2020

PCOMPBIOL-D-19-02054R1 

Insights into adenosine A2A receptor activation through cooperative modulation of agonist and allosteric lipid interactions

Dear Dr Giraldo,

I am pleased to inform you that your manuscript has been formally accepted for publication in PLOS Computational Biology. Your manuscript is now with our production department and you will be notified of the publication date in due course.

With kind regards,

Matt Lyles
